# Global, asynchronous partial sweeps at multiple insecticide resistance genes in *Aedes* mosquitoes

Thomas L. Schmidt [1] ✉, Nancy M. Endersby-Harshman[1], Anthony R. J. van Rooyen[2], Michelle Katusele [3], Rebecca Vinit[3], Leanne J. Robinson[3,4], Moses Laman[3], Stephan Karl [4,5] & Ary A. Hoffmann [1]

*Aedes aegypti* (yellow fever mosquito) and *Ae. albopictus* (Asian tiger mosquito) are globally invasive pests that confer the world's dengue burden. Insecticide-based management has led to the evolution of insecticide resistance in both species, though the genetic architecture and geographical spread of resistance remains incompletely understood. This study investigates partial selective sweeps at resistance genes on two chromosomes and characterises their spread across populations. Sweeps at the voltage-sensitive sodium channel (*VSSC*) gene on chromosome 3 correspond to one resistance-associated nucleotide substitution in *Ae. albopictus* and three in *Ae. aegypti*, including two substitutions at the same nucleotide position (F1534C) that have evolved and spread independently. In *Ae. aegypti*, we also identify partial sweeps at a second locus on chromosome 2. This locus contains 15 glutathione S-transferase (*GST*) epsilon class genes with significant copy number variation among populations and where three distinct genetic backgrounds have spread across the Indo-Pacific region, the Americas, and Australia. Local geographical patterns and linkage networks indicate *VSSC* and *GST* backgrounds probably spread at different times and interact locally with different genes to produce resistance phenotypes. These findings highlight the rapid global spread of resistance and are evidence for the critical importance of *GST* genes in resistance evolution.

Insecticide resistance (hereafter: 'resistance') in mosquitoes and other insect pests remains one of the most pressing issues in global public health and food security, and the mechanisms underpinning its evolution continue to be a major focus of research[1,2]. Genes associated with resistance include those conferring target-site resistance (e.g., mutations in the voltage-sensitive sodium channel (*VSSC*) gene, also known as *kdr* mutations[3]) or metabolic resistance (e.g., glutathione S-transferase (*GST*)[1] and cytochrome P450 (*P450* or *CYP*) genes[4]). Genomic research on pests has helped identify how different resistance profiles are produced and maintained, such as by linkage, epistasis, and structural variation[5–8], and can indicate how specific types of resistance become geographically widespread via gene flow between populations[9] or species[10] or by multiple independent substitutions of the same allele[11,12]. As resistance spreads, it can produce striking patterns of geographical genetic structure in response to local selection pressures[13]; this 'local' environment is determined by local insecticide usage, and thus geographically distant populations may have identical resistance alleles while proximate populations do not[9,14].

[1]Bio21 Institute, School of BioSciences, University of Melbourne, Parkville, Australia. [2]Cesar Australia, Brunswick, Victoria, Australia. [3]PNG Institute of Medical Research, Madang, Madang Province, Papua New Guinea. [4]Australian Institute of Tropical Health and Medicine, James Cook University, Smithfield, Queensland, Australia. [5]Burnet Institute of Medical Research, Melbourne, Victoria, Australia. ✉e-mail: toms@unimelb.edu.au

Genetic structure at resistance genes can contrast sharply with genome-wide structure, which in invasive pests with large populations is generally established and reinforced by demographic processes during and after colonisation[15].

*Aedes aegypti* (yellow fever mosquito) and *Ae. albopictus* (Asian tiger mosquito) are highly invasive pests with a global dengue burden of ~390 million infections per year[16]. Both species spread by human conveyance and have colonised much of the world's tropics and subtropics and some temperate regions[17,18]. In recent decades, incidental human movement has helped spread resistance alleles between populations of both species[9,12,19,20]. For *Ae. aegypti*, most of this spread will have taken place between invasive populations that established between the 16th and 19th centuries following an initial expansion from the African native range[21]. The invasion history of *Ae. albopictus* before the 20th century is less well understood, though the species has recently spread from its native range in Asia to colonise the Americas, the Pacific region, and Europe[22], and this current range expansion may thus also be spreading resistance alleles into new areas.

Evolutionary investigations of resistance in *Aedes* have predominantly focused on the *VSSC* gene, as resistant phenotypes can be traced to variation in this single gene. Several point mutations at the *VSSC* gene have been associated with resistance, but F1534C and V1016G are the most well-characterised and widely distributed. F1534C has a global distribution in both species[9,12], while V1016G is common among *Ae. aegypti* in the Indo-Pacific region where it may be locally fixed or segregating alongside F1534C[9]. V1016G has also recently been reported in *Ae. albopictus*[23]. F1534C and V1016G confer strong resistance to Type I pyrethroid insecticides (e.g., permethrin), while V1016G also resists Type II pyrethroids (e.g., deltamethrin)[3]. The advantages of *VSSC* mutations appear to be offset by physiological costs[24], which should favour the wild-type susceptible background when insecticide use is low. This trade-off has been observed in *Culex* mosquitoes, where local frequencies of resistant and wild-type backgrounds rise and fall in response to seasonal insecticide usage[13]. In *Aedes* mosquitoes, this trade-off may also explain why wild-type mosquitoes can remain common in populations where *VSSC* resistance alleles are present[9] and why these alleles remain absent in some regions with low insecticide usage[25].

Genomic studies of *Ae. aegypti* have found that F1534C has likely reached its distribution via two independent substitutions which have each spread globally[12,19]. The distribution of V1016G appears to be from a single substitution and is usually observed with a third mutation, S989P, but it is rarely observed on the same haplotype as F1534C[9]. These alleles spread across populations by positive selection, which produces a selective sweep pattern in which individuals sharing the same allele will also be more genetically homogeneous at sites near the sweep locus than they are at other regions of the genome[26]. A previous genomic study found selective sweeps through *Ae. aegypti* populations had structured genetic variation at sites more than 10 Mb from the *VSSC* gene[20].

Genomic regions undergoing selective sweeps typically display three key evolutionary patterns: an increase in linkage disequilibrium, an increase in the proportion of alleles at low-frequency, and an increase in population genetic differentiation[27]. Within a single sample of individuals, these increases can be identified relative to other genomic regions. If sweeps are incomplete or partial, they can still be more powerfully analysed using a second sample with a different swept background or no selection history at the locus of interest[28]. If two different genetic backgrounds have undergone independent sweeps at the same locus, the two samples should be highly differentiated at the locus and this locus should show similarly high levels of linkage disequilibrium and rare allele proportions in each swept background. When one sample has experienced a sweep, and one has not, these should be differentiated at the sweep locus, and this locus should have higher levels of linkage disequilibrium and rare allele

proportions in the sweep sample relative to the other sample. These patterns should be particularly stark if the sweep extends over populations strongly differentiated at other genomic regions.

This study analyses global patterns of genome-wide genetic structure in *Ae. aegypti* and *Ae. albopictus*, and detects SNPs that are structured in line with genotypes at *VSSC* point mutations rather than genome-wide structure. We use these SNPs to confirm two evolutionarily independent, partial sweeps of F1534C and one of V1016G in *Ae. aegypti*, as well as a single sweep of F1534C in *Ae. albopictus*; these findings expand current geographical knowledge of these sweeps and show how these have spread across distant populations. However, our analysis also detected another genomic region in *Ae. aegypti* which has undergone three evolutionarily independent, partial selective sweeps, each segregating throughout multiple populations. Comparisons with other individuals from these populations showed the sweep-associated individuals had increased linkage disequilibrium, a greater proportion of low-frequency alleles, and stronger differentiation around the sweep locus, with similar patterns to those at the *VSSC* gene. This second sweep locus contained 15 *GST* epsilon class genes that had significant variation in gene copy number across countries. While *GST* epsilon genes have been linked to resistance to insecticides including DDT, organophosphates, and pyrethroids in *Aedes* and other mosquitoes[29–31], they have often been operationally treated as having complementary contributions to metabolic resistance phenotypes alongside the many esterase and mono-oxygenase genes as well as other *GST*s[32]. The three evolutionarily independent, globally segregating backgrounds at the *GST* epsilon genes instead suggest these genes have a critical role in resistance to chemical control of *Ae. aegypti*.

## Results

### Microhaplotype differentiation reveals global genetic structure and admixture patterns

Prior to investigating patterns of gene flow at specific genomic loci, we first investigated patterns of genetic structure and admixture across the genome. For this we used double digest restriction-site associated DNA sequencing (ddRADseq) data from 934 *Ae. aegypti* and *Ae. albopictus* individuals, including 358 sequenced for this study. This included *Ae. aegypti* from 19 countries ($n = 444$ and Supplementary Data 1) and *Ae. albopictus* from 17 countries ($n = 490$ and Supplementary Data 2).

As ddRAD data are sequenced as microhaplotypes, we investigated global genomic structure from patterns of haplotype coancestry, using fineRADstructure[33]. This uses the sequence of all the SNPs from each RAD locus to find one or more closest relatives for each microhaplotype, producing coancestry matrices of individual haplotype similarity for *Ae. aegypti* (Fig. 1a; 79,084 SNPs) and *Ae. albopictus* (Fig. 2a; 96,269 SNPs). For both species, individuals formed clusters based on their population of origin and these populations tended to cluster by their geographical location. Notable exceptions were recent invasions of *Ae. albopictus* which clustered with their previously-identified source populations; these included Fiji (invaded from Southeast Asia[34]), Mauritius (invaded from East Asia[34]), and the Torres Strait Islands (invaded from Indonesia[35]) (Fig. 2a). Figure 2a also indicates that the recent *Ae. albopictus* invasion of Vanuatu originated in Papua New Guinea (PNG) or nearby.

Putatively admixed populations were identified in *Ae. aegypti* from New Caledonia and *Ae. albopictus* from the North Fly Region, PNG, which clustered apart from other populations in their regions (Figs. 1a, 2a). We explored these relationships further using TreeMix[36] (Figs. 1b, 2b), which builds maximum likelihood trees from population allele frequencies and then adds optional migration edges between populations. We ran TreeMix on subsets of populations selected from the fineRADstructure analysis, including putatively admixed populations, populations from potential source clades, and populations from

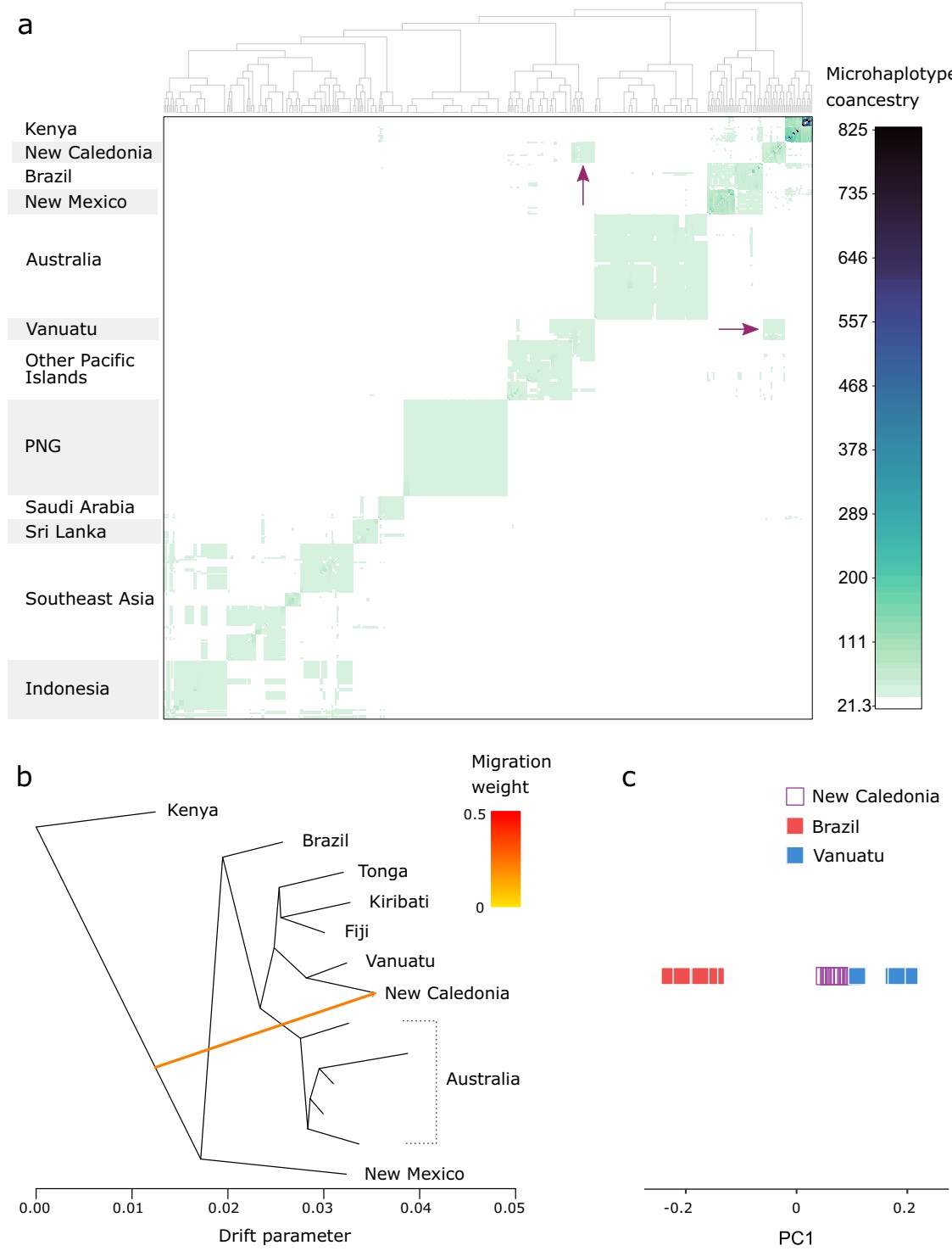

**Fig. 1 | Genetic structure and admixture in *Ae. aegypti*. a** Analysis of haplotype similarities among individuals in fineRADstructure[33]. Darker colours indicate higher pairwise similarity of RADtag microhaplotypes. Arrows indicate putative admixture. **b** Maximum likelihood tree assessing admixture in TreeMix[36]. TreeMix was run on windows of 500 SNPs using a subset of populations associated with the putative admixture identifed in (**a**). The coloured arrow indicates the migration edge. The single best tree of 100 is reported. **c** Principal components analysis of the putative admixed population and source populations. *Aedes aegypti* from New Caledonia are projected onto a single principal component constructed from variation between the two putative source populations. The position of the admixed individuals relative to the source populations reflects the admixture proportions[39].

clades intermediate to these. We assessed migration edge placement using 100 runs of TreeMix with either zero, one, or two migration edges added to the maximum likelihood trees. When TreeMix was run with one migration edge, these were placed in the same spots in all 100 TreeMix runs, and these trees had a higher likelihood than the zero-

migration trees in 98 (*Ae. aegypti*) and 99 (*Ae. albopictus*) of the 100 runs. When TreeMix was run with two migration edges, these were inconsistently placed across runs, so we used the tree with the highest likelihood and one migration edge. Here, *Ae. aegypti* from New Caledonia appeared to be an admixture of a population in the Americas or

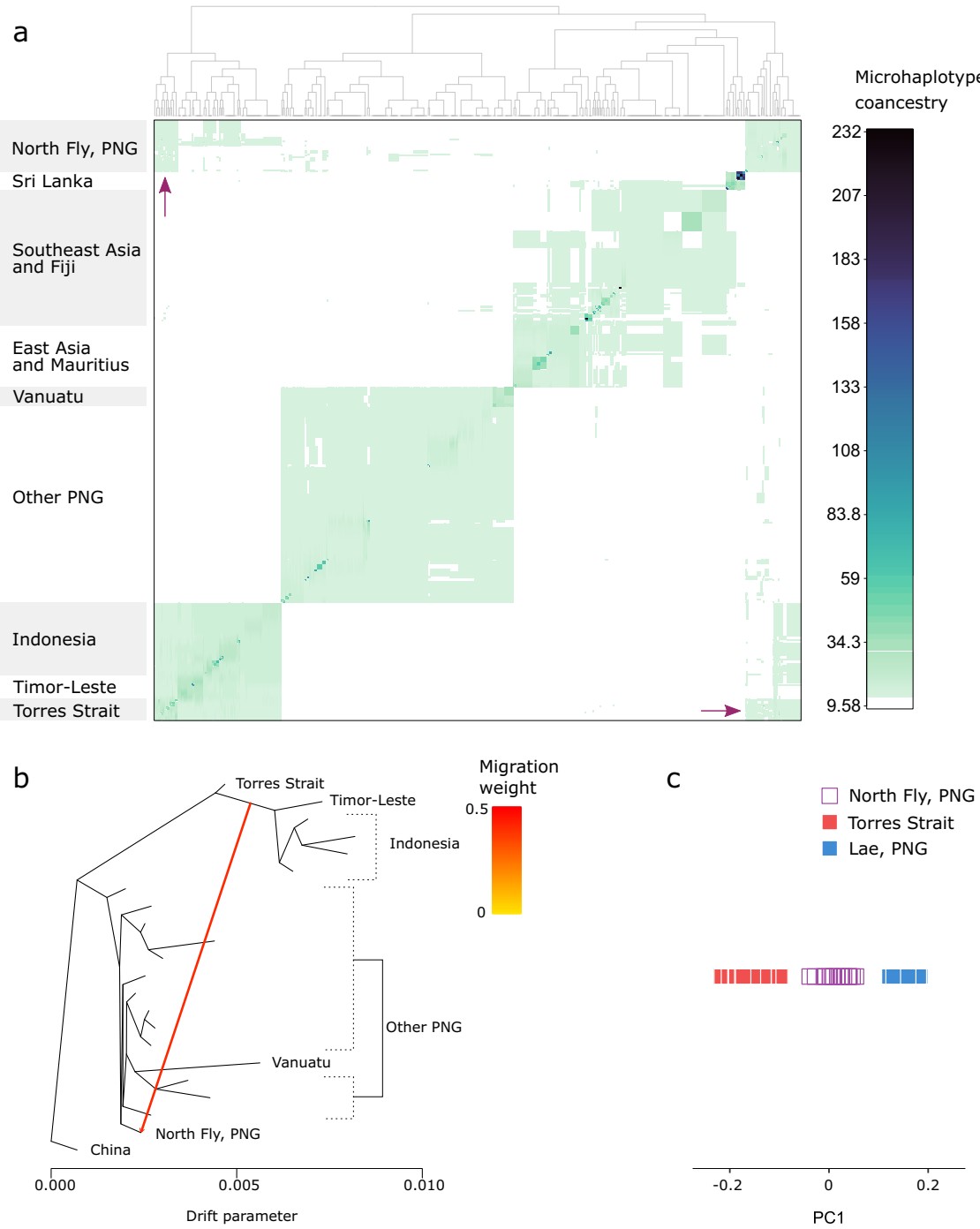

**Fig. 2 | Genetic structure and admixture in *Ae. albopictus*. a** Analysis of haplotype similarities among individuals in fineRADstructure[33]. Darker colours indicate higher pairwise similarity of RADtag microhaplotypes. Arrows indicate putative admixture. **b** Maximum likelihood tree assessing admixture in TreeMix[36]. TreeMix was run on windows of 500 SNPs using a subset of populations associated with the putative admixture identified in (**a**). The coloured arrow indicates the migration edge. The single best tree of 100 is reported. **c** Principal components analysis of the putative admixed population and source populations. *Aedes albopictus* from North Fly, PNG, are projected onto a single principal component constructed from variation between the two putative source populations. The position of the admixed individuals relative to the source populations reflects the admixture proportions[39].

Africa (most likely the Americas given Fig. 1a) and a population from the Pacific Islands (most likely Vanuatu) (Fig. 1b), while *Ae. albopictus* from the North Fly Region, PNG, was an admixture between other PNG populations and one or more populations from Indonesia, Timor-Leste, and the Torres Strait Islands (Fig. 2b). We investigated these further using $F_3$ statistics on all population triads. F3 statistics test whether one population has been produced from the admixture of two others and are particularly powerful for detecting recent admixture

events that involve equal proportions from each population[37], though they may be underpowered for reduced representation data[38]. $F_3$ tests were statistically significant for admixture in the North Fly Region, sourced from any of the other PNG populations and either the Torres Strait Islands or any Indonesian population ($-3.95 \times 10^{-4}$ <All $F_3 < -1.30 \times 10^{-3}$, 0.031 > All Bonferroni *P*-values > $1 \times 10^{-23}$). The most statistically probable pair of source populations were Bali, Indonesia and Madang, PNG. No $F_3$ tests were significant for any *Ae. aegypti*

populations, including comparisons involving New Caledonia. This result may reflect limitations of $F_3$ tests in detecting older or asymmetrical admixture[37], which may be the case in New Caledonia and may also reflect our limited sampling of American *Ae. aegypti* populations.

When admixed individuals are projected onto a principal component axis constructed from individuals from the two source populations, the position of the admixed individuals relative to the source population individuals reflects the admixture proportions[39]. These PCAs indicated that New Caledonian *Ae. aegypti* were much more similar to Vanuatu than Brazil (Fig. 1c), while *Ae. albopictus* from the North Fly Region, PNG, had equivalent proportions of Torres Strait Island and PNG backgrounds (Fig. 2c).

### Three globally-distributed partial sweeps at the voltage-sensitive sodium channel (*VSSC*) in *Aedes aegypti*

To investigate selective sweeps, we aimed to identify genetic backgrounds at the *VSSC* gene that were common among geographically distinct populations of *Aedes* mosquitoes and that showed signs of positive selection. For this, we supplemented the genome-wide sequence data with endpoint genotyping assays that scored *Ae. aegypti* for three mutations at the *VSSC* gene (V1016G, F1534C, and S989P) and *Ae. albopictus* for two (V1016G and F1534C). In *Ae. aegypti*, all three mutations were detected, but as S989P was only found in the presence of V1016G, we did not analyse it further. Genotypes at this site are recorded in Supplementary Data 1. The V1016G and F1534C mutations were both geographically widespread and had similar global frequencies (0.333 and 0.327, respectively), though V1016G was not observed outside the Indo-Pacific region, and neither mutation was found in any Australian population (Fig. 3a). In Ae. albopictus, we did not detect any individuals with V1016G. After removing samples with uncertain *VSSC* genotypes, this left 395 *Ae. aegypti* and 490 *Ae. albopictus* for analysis. The frequencies of each allele in each population are listed in Supplementary Data 3, 4.

For *Ae. aegypti*, we first used latent factor mixed models to identify SNPs where the genetic structure was segregating in line with genotypes at V1016G and F1534C rather than with genome-wide genetic structure. Genome-wide genetic structure was conditioned for in each mixed model via sparse non-negative matrix factorisation on the 51,115 genome-wide SNPs retained after filtering (Supplementary Fig. 1), setting $K = 18$. Outlier SNPs were identified as those with adjusted P-values below the inverse of the number of SNPs ($1.956 \times 10^{-5}$). We used these SNPs to identify genetic backgrounds shared across distinct populations and assessed these for evidence of multiple independent substitutions of the same resistance mutation.

Mixed models identified 59 and 47 SNPs in and around the *VSSC* gene that were strongly structured in line with V1016G (Fig. 3b) and F1534C (Fig. 3c). Strongly associated SNPs are listed in Supplementary Data 5 and covered a region 2,555,236 bp overlapping the *VSSC* gene. Forty-two SNPs were common to both models (Fig. 4a). Curiously, a second genomic region at ~ 351 Mb on chromosome 2 also contained 13 (V1016G, Fig. 3b) and 19 (F1534C, Fig. 3c) SNPs strongly structured by *VSSC* genotype. This region contained 15 glutathione S-transferase (*GST*) genes and is analysed in section "Three partial sweeps and copy number variation at glutathione S-transferase epsilon genes (*GSTs*) in *Aedes aegypti*" Associations of SNPs with this second region appeared to be driven by the *VSSC* wild-type individuals from Australia, and rerunning the mixed models with these removed found no associations with the *GST* region, though the F1534C mutation was associated with a 'Nach' sodium channel protein (Supplementary Fig. 2).

We used the PCA of the 42 SNPs common to both models to infer evolutionary patterns (Fig. 4b). V1016G homozygotes formed a single main cluster ($n = 98$), though some were spread toward the centre of the PCA likely as a result of missing data or recombination. F1534C homozygotes formed two main clusters. One of these included individuals from North and South America, New Caledonia, Saudi Arabia,

and Malaysia ($n = 21$; Fig. 4c; dark red) and the other included Kenya, Thailand, Malaysia, Fiji, Tonga, and Kiribati ($n = 43$; Fig. 4c; orange).

We tested the hypothesis that the V1016G cluster and two F1534C clusters represented three independent sweeps spread across multiple populations via positive selection. To do this, we first estimated Tajima's D (D), nucleotide diversity (π), and linkage disequilibrium (LD, taken as $r^2$, the squared correlation coefficient between genotypes) between homozygotes assigned to each sweep (Fig. 4c) and calculated the difference between these parameter estimates and those from all *VSSC* wild-type individuals ($n = 113$). In each case, the region around the *VSSC* gene showed unique patterns for ΔD (Fig. 5a), Δπ (Fig. 5b), and ΔLD (Fig. 5c). For each putative sweep, ΔD and Δπ were lowest at the *VSSC* locus, while ΔLD was higher and with a broader peak than other regions on chromosome 3. The V1016G sweep had the weakest LD signal (Fig. 5c), possibly due to the inclusion of recombined individuals (Fig. 4b).

We calculated genome-wide pairwise $F_{ST}$ between each pair of *VSSC* backgrounds to investigate these patterns further. Sharp $F_{ST}$ peaks at the *VSSC* gene reflected many fixed differences between the three backgrounds (Fig. 5d; 'v'), suggesting these represent independent evolutionary events that have swept different alleles towards fixation. These analyses also indicated several other peaks across the genome (Fig. 5d; 'i–iv') containing genes with functions relevant to insecticide resistance (complete list in Supplementary Data 6). These include (i) four ionotropic glutamate receptor genes with sodium transport function linked to insecticide resistance[40], and a probable glutathione peroxidase 2 gene with oxidative stress function linked to insecticide resistance[41]; (ii) a sodium/potassium-transporting ATPase gene with sodium transport function, and a cytochrome P450 gene (*CYP6BB2*) with insecticide resistance function[42]; (iii) a glutamate-gated chloride channel gene with insecticide resistance function[43]; and (iv) an ankyrin gene with a function associated with the *VSSC* gene[44,45]. The regions 'i–iv' with high $F_{ST}$ were specific to comparisons between the F1534C Americas background (dark red) and the two Indo-Pacific backgrounds of F1534C (orange) and V1016G (teal).

### Three partial sweeps and copy number variation at glutathione S-transferase epsilon genes (*GSTs*) in *Aedes aegypti*

The region of SNPs on chromosome 2 strongly associated with *VSSC* genotype covered a 2,761,683 bp region. This region contained 32 SNPs strongly associated (adjusted *P*-values $< 1.956 \times 10^{-5}$) with the V1016G and F1534C mutations (Fig. 6a; 13 and 19 SNPs respectively). Genes within this region included 15 glutathione S-transferase (*GST*) epsilon-class genes with obvious links to resistance[29–31], as well as an ankyrin-3 gene with possible *VSSC*-associated function[44,45]. Positions of the 32 SNPs are listed in Supplementary Data 7.

Visual analysis of variation at these SNPs identified three homozygous haplotypes in multiple individuals from different populations (Fig. 6b). These were found in individuals from across all five Australian populations (GST Australia, purple, $n = 24$), from Thailand, Malaysia, Indonesia, and Papua New Guinea (PNG) (GST Indo-Pacific, blue, $n = 9$), and from Brazil, Kiribati, Fiji, New Caledonia, and Vanuatu (GST Americas, green, $n = 10$) (Fig. 6c). *VSSC* genotypes of these 43 individuals were diverse, including homozygotes of V1016G (GST Indo-Pacific and GST Americas), F1534C Indo-Pacific (GST Indo-Pacific and GST Americas), F1534C Americas (GST Americas), and wild type (GST Australia and GST Americas) (Fig. 6b). Although the three *GST* groups were identified by requiring all individuals to have homozygous genotypes at all 32 SNPs, relaxing this requirement to allow for up to one heterozygous or missing site added New Mexico and Port Moresby (PNG) to the GST Americas group. Port Moresby contained individuals from both the GST Americas and GST Indo-Pacific groups (Fig. 6c).

We tested the hypothesis that the three *GST* haplotypes represented three independent sweeps spread across multiple populations via positive selection. Here we followed the same approach as for *VSSC*

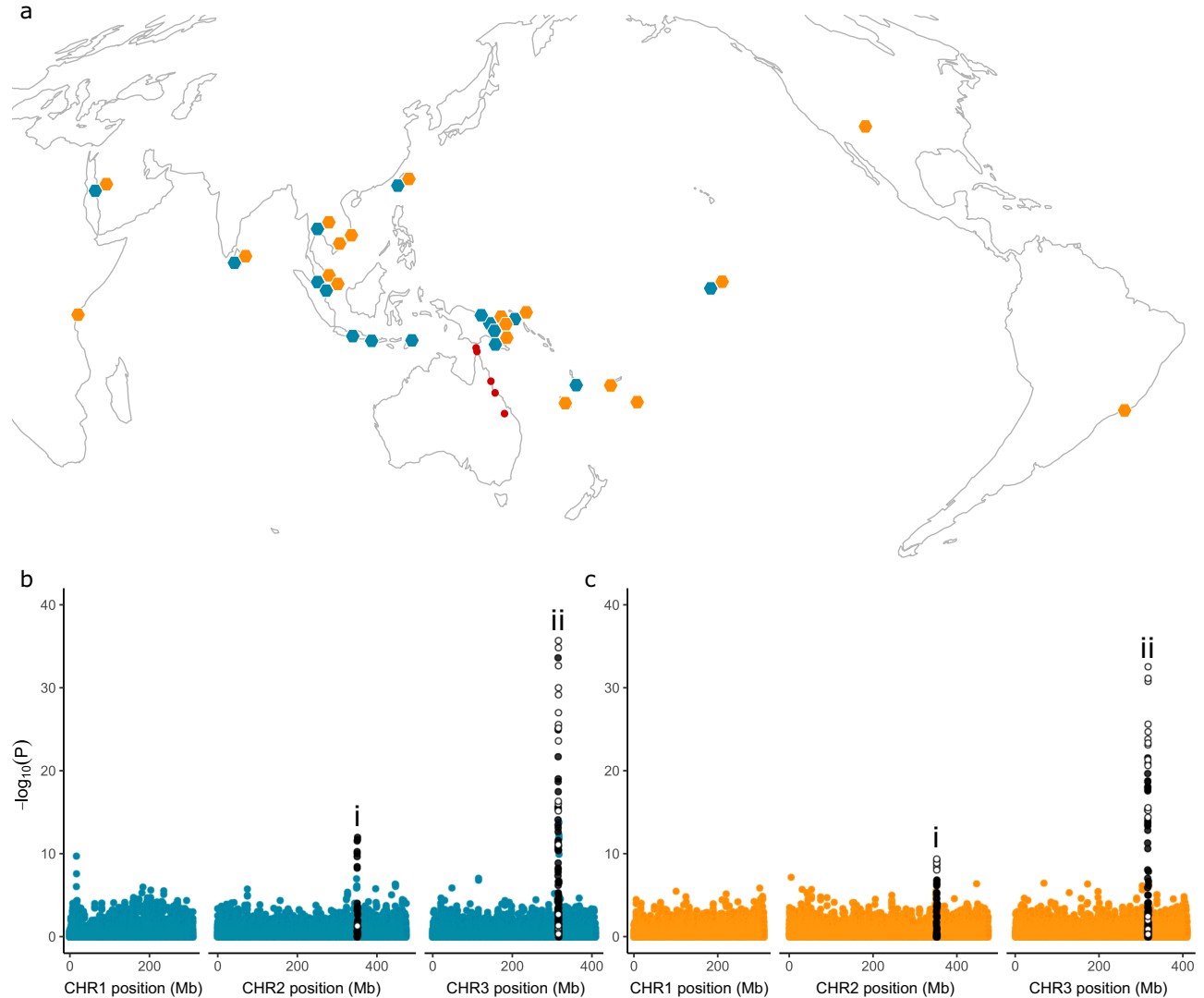

**Fig. 3 | Geographical distributions of voltage-sensitive sodium channel (*VSSC*) mutations and their association with genome-wide SNPs in *Ae. aegypti*.** **a** Coloured hexagons indicate populations containing the V1016G (teal) and F1534C (orange) mutations, with double hexagons indicating populations with both mutations. Red circles indicate populations where neither mutation was observed. Local frequencies of each mutation are listed in Supplementary Data 3. **b**, **c** Latent factor mixed models for V1016G (**b**) and F1534C (**c**). Mixed models identify genome-wide SNPs that have genetic structure in line with *VSSC* genotype after controlling for genome-wide patterns assessed using sparse non-negative matrix factorisation with $K = 18$ (Supplementary Fig. 1). White circles indicate SNPs within the *VSSC* gene on chromosome 3 ('ii') or the region containing 15 glutathione S-transferase (*GST*) genes on chromosome 2 ('i'), black circles indicate SNPs within 1 Mb of either region. *P*-values were adjusted from combined z-scores using a Benjamini-Hochberg correction with a false discovery rate of 0.01.

backgrounds (Fig. 4), but, as the *GST* haplotypes were at lower frequencies than the *VSSC* mutations, there were enough wild types for us to restrict the ΔD, Δπ, and ΔLD comparisons to the same sets of populations as the sweeps: $n = 47$ (GST Australia), $n = 89$ (GST Indo-Pacific), and $n = 62$ (GST Americas). This should provide much more substantial evidence that the *GST* haplotypes were indicators of sweep backgrounds, and indicate whether the putative sweep backgrounds were segregating in populations (i.e., partial sweeps). In each case, the region around the *GST* genes showed unique patterns for ΔD (Fig. 7a), Δπ (Fig. 7b), and ΔLD (Fig. 7c) that were similar to the *VSSC* gene. For each putative sweep, ΔD and Δπ were lowest at the *GST* locus, while ΔLD had a much broader peak than other regions on chromosome 2. Pairwise $F_{ST}$ between each sweep background showed sharp peaks at the *GST* region in line with the differential fixation of alleles (Fig. 7d; 'ii'). A second peak was also apparent on chromosome 1 in the two comparisons with the GST Australia haplotype (Fig. 7d; 'i'). This contained a glutathione synthetase gene with antioxidant functions linked to insecticide resistance[46] (Supplementary Data 8). Together these

results suggest that the three haplotypes represent evolutionarily independent partial sweeps segregating in their populations.

Copy number variation among individuals was inferred through the ratio of read depths at sites within the coding regions of the 15 *GST* genes relative to the average depth at sites 10 Mb upstream and downstream from the *GST* region. Individuals with depth recorded at fewer than 500 sites within *GST* coding regions were omitted. For all individuals, downstream read depths were similar to upstream read depths ($R^2 = 0.98$). Read depth ratios varied strongly by geographical region (ANOVA, $F_{9,263} = 60.0$, $P = 1.1e^{-58}$) and were on average lower than upstream and downstream ($\bar{x} = 0.86 \times$) (Fig. 8a). Ratios were much higher for individuals from North America (New Mexico, $\bar{x} = 2.62 \times$) but lower in the Middle East (Saudi Arabia, $\bar{x} = 0.31 \times$) and South Asia (Sri Lanka, $\bar{x} = 0.30 \times$); these two latter populations are genomically similar (Fig. 1a). The higher read depths in New Mexico are unlikely to be from laboratory or sequencing errors, as these individuals were prepared and sequenced in the same DNA library as East Africa (Kenya). Read depth ratios showed no difference between

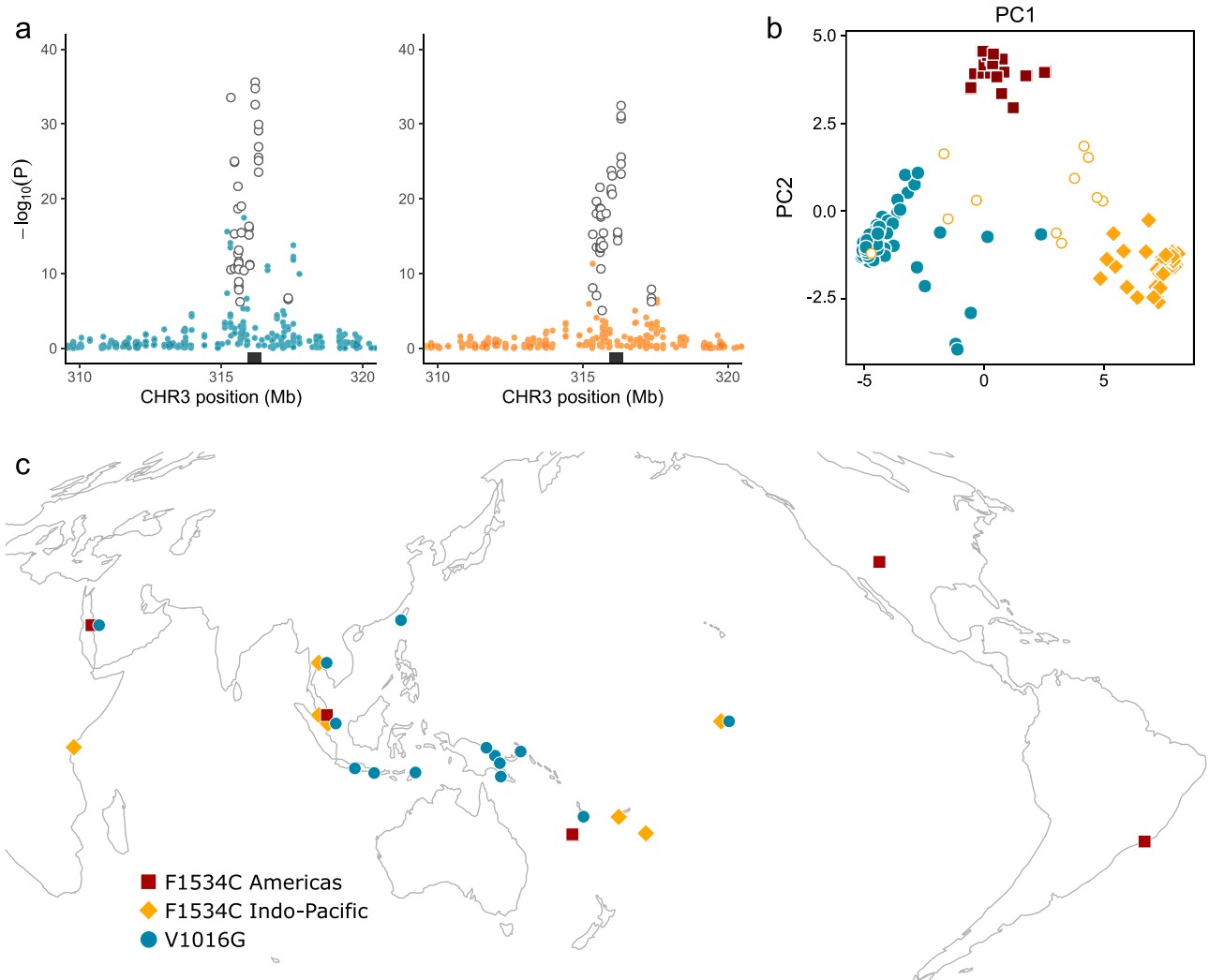

**Fig. 4 | Identification of three sweep backgrounds at the voltage-sensitive sodium channel (*VSSC*) gene in *Ae. aegypti*. a** Local detail of latent factor mixed model results around the *VSSC* gene region. Data and statistical tests are identical to Fig. 3b, c. Black rectangles indicate the *VSSC* gene region. White-filled circles indicate 42 SNPs identified by both models beyond the $P < 1.956 \times 10^{-5}$ significance cut-off. **b** PCA of all *VSSC* homozygotes, using the 42 outlier SNPs. Symbols of solid colour indicate homozygotes of the three swept backgrounds. White-filled circles indicate F1534C homozygotes of uncertain background. **c** Locations containing at least one *Ae. aegypti* homozygote assigned to one of the three *VSSC* sweeps. Colours and shapes indicate the three swept backgrounds.

individuals with *GST* sweep haplotypes and wild-type individuals from the same regions (Fig. 8b). No signs of copy number variation were observed at the *VSSC* gene, though V1016G homozygotes had slightly higher relative read depths (Supplementary Fig. 3).

**Linkage disequilibrium networks in *Aedes aegypti***
Linkage disequilibrium networks can provide useful insights into the genetic architecture of local adaptation[47]. For each of the six sweep backgrounds in *Ae. aegypti* (three *VSSC*, three *GST*), we identified and investigated other regions of the genome that were in strong linkage (r²) with the sweep locus. As before, we analysed individuals in subsets corresponding to each sweep background and wild types. We considered only SNPs that were at least 50 Mb from the sweep locus and with r² > 0.6 to at least one SNP within 1 Mb of the locus. SNPs were scored for each r² > 0.6 interaction with a SNP near the locus, and these were plotted as histograms using 500 kb bins (Fig. 9).

Peaks were observed for each of the six sweep backgrounds, and these overlapped genes with products linked to insecticide resistance, including cytochrome P450[4], glutathione S-transferase[1], acetylcholinesterase[6], cytochrome b5[48], cytochrome c oxidase[49], esterase B1[50], Nach sodium channel protein[3], cholinesterase[51], UDP-

glycosyltransferase[52], carbonic anhydrase II[53], trypsin and anti-chymotrypsin[54], peroxiredoxin[55], and sodium-coupled mono-carboxylate transporter[56]. Both F1534C backgrounds were strongly linked to the *GST* epsilon gene region, but no *GST* backgrounds were linked to the *VSSC* region. Several other genes were linked to multiple sweeps. These included a cluster of 19 cytochrome P450 genes on chromosome 2, a second cluster of 3 cytochrome P450 genes and a Nach sodium channel protein gene on chromosome 3, an acet-ylcholinesterase gene on chromosome 3, and an anti-chymotrypsin gene on chromosome 3 (Fig. 9). GST Australia had the fewest peaks associated with resistance genes. Wild-type individuals had far fewer strongly linked SNPs, and these formed smaller peaks at different locations to the sweep backgrounds (Supplementary Figs. 4, 5). We repeated the linkage network analyses on subsets of individuals from each background, either omitting countries or considering only spe-cific countries (Supplementary Figs. 6–10). These analyses indicated considerable heterogeneity in genetic architecture across countries and regions, where some genes were only associated with linkage peaks in specific regions and other genes were only associated with linkage peaks in the whole dataset. Genes identified through linkage network analysis are listed in Supplementary Data 9.

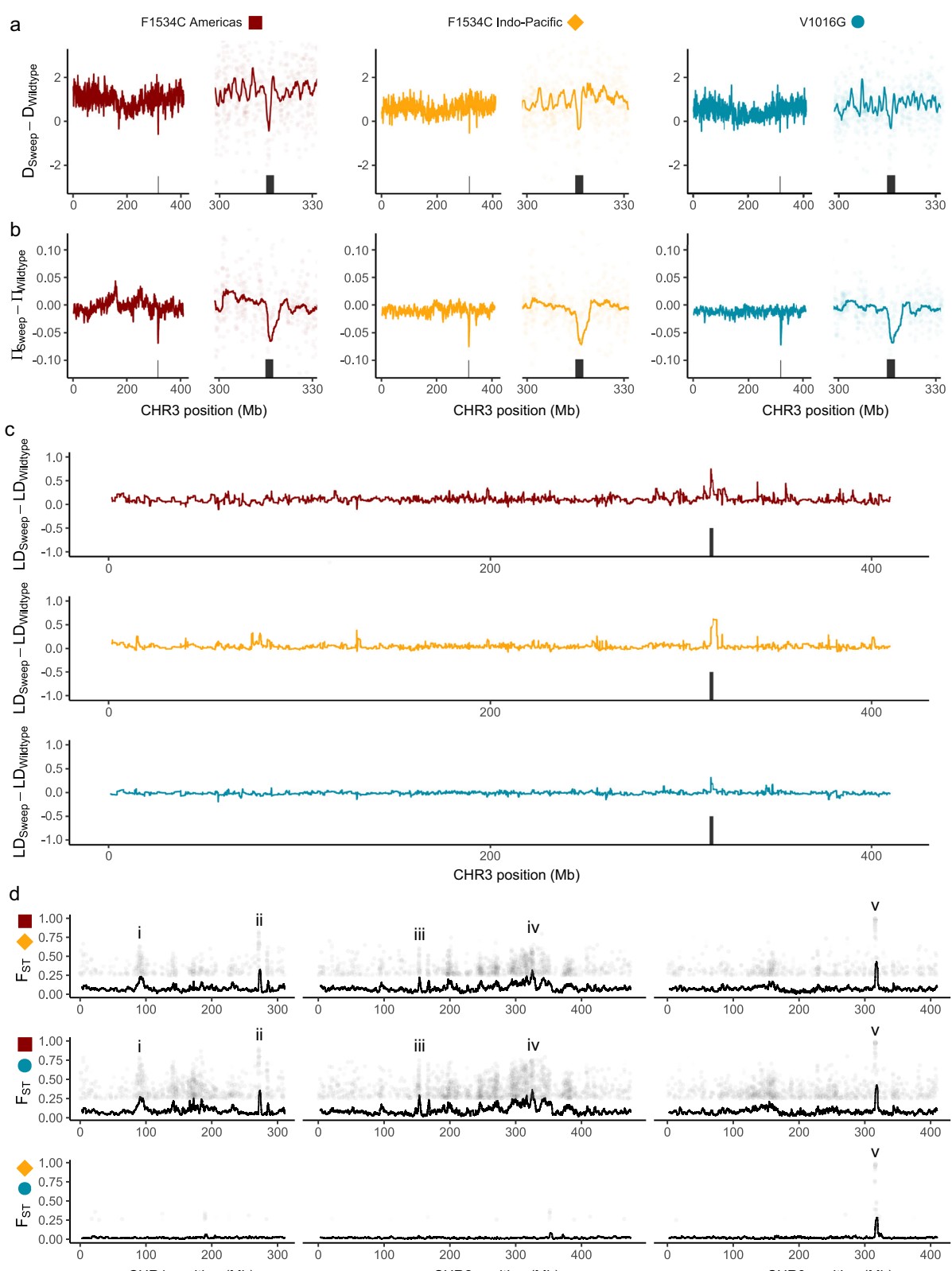

**Fig. 5 | Genomic characterisation of three sweeps at the voltage-sensitive sodium channel (*VSSC*) gene in *Ae. aegypti*. a** Difference in Tajima's D between sweep-associated individuals and wild-type individuals from all populations. Left-hand plots show all of chromosome 3, and right-hand plots focus on the gene region. Black rectangles indicate the *VSSC* gene region ±1 Mbp. Coloured lines show moving averages. **b** As above but showing differences in nucleotide diversity. **c** Moving average of the squared correlation coefficient between genotypes across chromosome 3. **d** Pairwise $F_{ST}$ between individuals of each swept background. Colours and shapes indicate the two swept backgrounds being compared. Black lines show moving averages, and plotted points show SNPs with $F_{ST} > 0.25$. Highly divergent regions of interest are denoted 'i'–'v' and described in the main text.

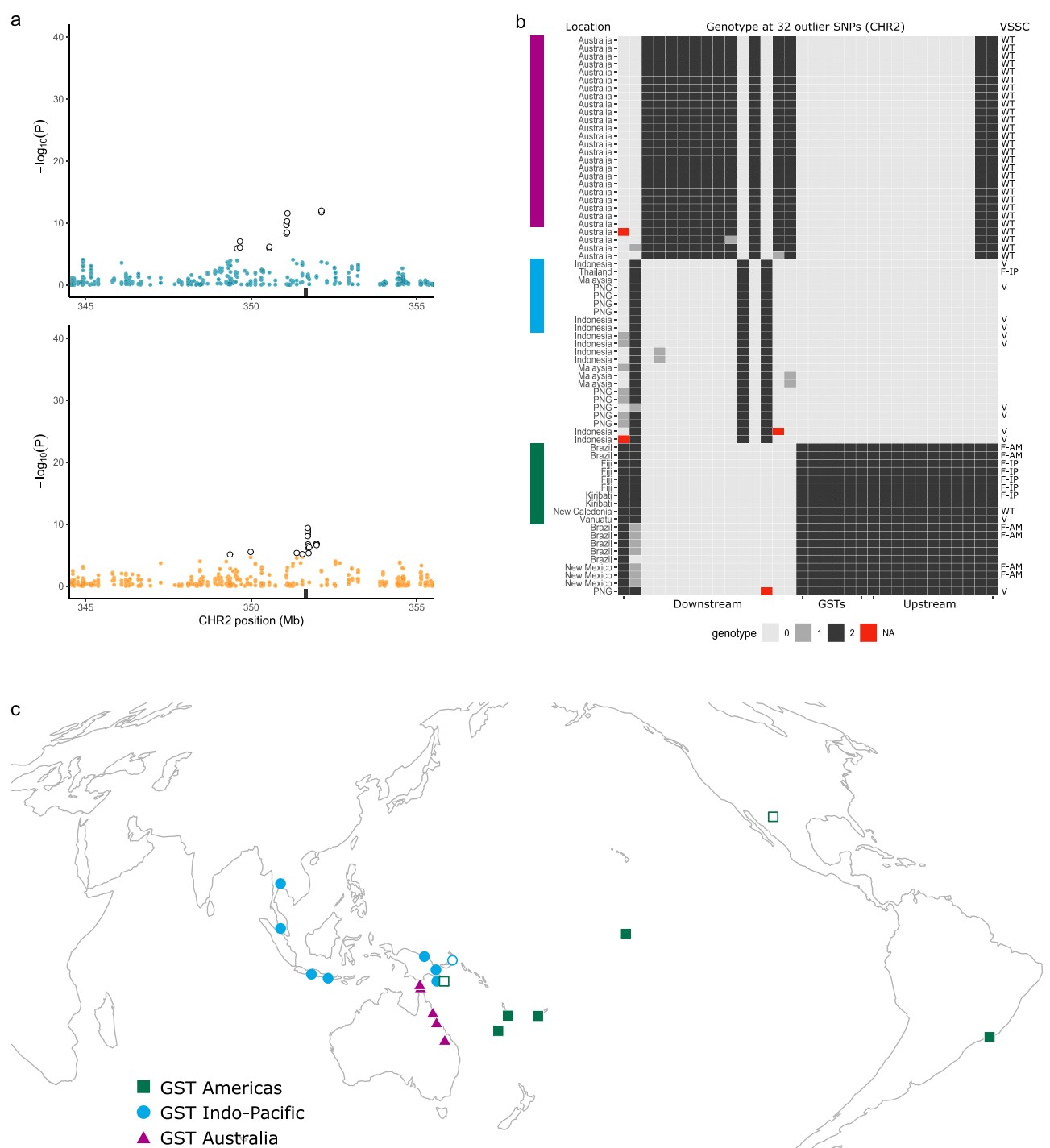

**Fig. 6 | Identification of three sweep haplotypes at 15 glutathione S-transferase (*GST*) genes in *Ae. aegypti*. a** Local detail of latent factor mixed model results around the *GST* gene region. Data and statistical tests are identical to Fig. 3b,c. Black rectangles indicate the region of 15 *GST* genes. White-filled circles indicate 32 SNPs beyond the $P < 1.956 \times 10^{-5}$ significance cut-off. **b** Genotypes at the 32 outlier SNPs identified through latent factor mixed models. Each row represents one individual mosquito, with genotypes shaded by the number of copies of the reference allele. Individuals indicated by the coloured rectangles have two copies of the same 32 SNP haplotype, other plotted individuals had up to one missing or heterozygous site. No other individuals are plotted. SNPs are split into those downstream, upstream, and within the region containing *GST* genes. The left-hand side y-axis lists the population of origin. The right-hand side y-axis indicates the VSSC genotype (WT = wild-type; V = V1016G background; F-A = F1534C Americas background; F-IP = F1534C Indo-Pacific background). **c** Locations of *Ae. aegypti* with two copies of a 32 SNP haplotype (coloured fill) or with one missing or heterozygous site (white fill). Colours indicate the three backgrounds following (**b**).

## A single partial sweep at the voltage-sensitive sodium channel (*VSSC*) in *Aedes albopictus*

We screened *Aedes albopictus* for the V1016G and F1534C mutations, finding no evidence of V1016G but six populations with F1534C (Fig. 10a and Supplementary Data 4). These were Singapore, Timor-

Leste, Vanuatu, and three PNG populations. Frequencies of F1534C (0.033) were ten times lower than in *Ae. aegypti* (0.327), with only a single F1534C homozygote detected in PNG, and the North Fly PNG population had only a single heterozygous individual ($n = 42$ assayed; frequency = 0.012). We investigated signs of positively selected

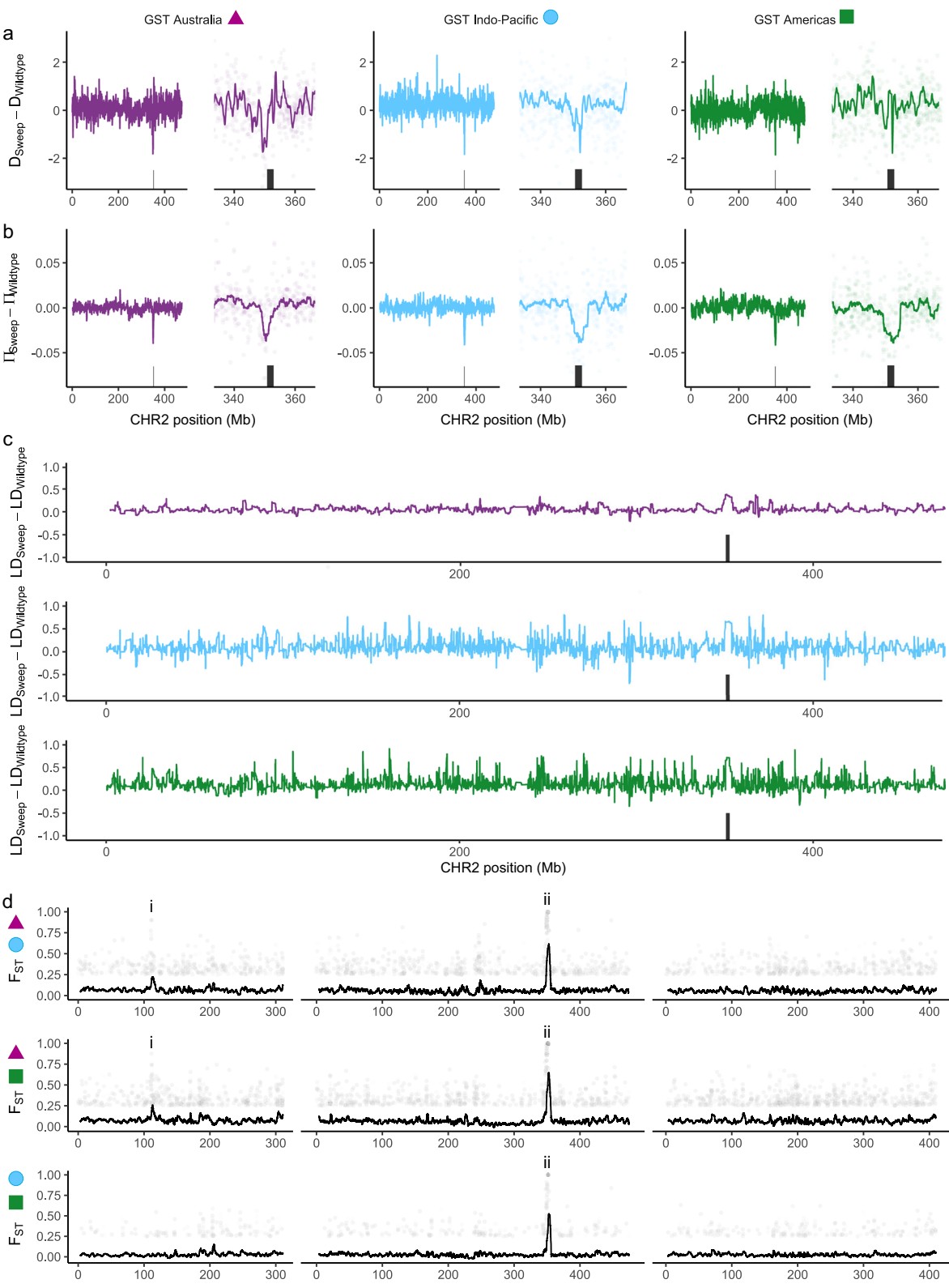

**Fig. 7 | Genomic characterisation of three sweeps at 15 glutathione S-transferase (*GST*) genes in *Ae. aegypti*. a** Difference in Tajima's D between sweep-associated individuals and wild-type individuals from the same populations as the sweeps. Left-hand plots show all of chromosome 2, and right-hand plots focus on the gene region. Black rectangles indicate the *GST* gene region ± 1 Mbp. Coloured lines show moving averages. **b** As above but showing differences in nucleotide diversity. **c** Moving average of the squared correlation coefficient between genotypes across chromosome 2. **d** Pairwise $F_{ST}$ between individuals of each swept background. Colours and shapes indicate the two swept backgrounds being compared. Black lines show moving averages, and plotted points show SNPs with $F_{ST} > 0.25$. The *GST* region is indicated with 'ii', and the location of a glutathione synthetase gene is indicated with 'i'.

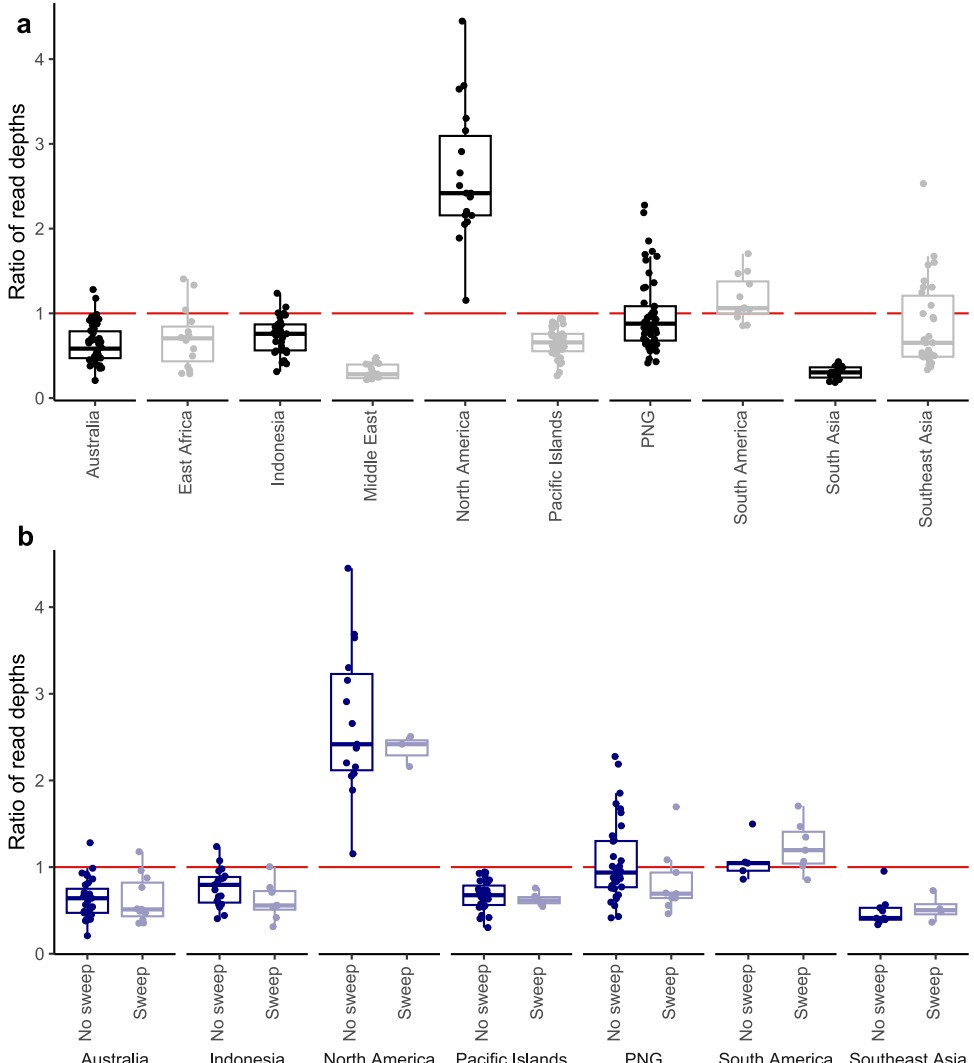

**Fig. 8 | Copy number variation at 15 glutathione S-transferase (*GST*) genes in *Ae. aegypti*.** *Y*-axis displays the ratio of reading depths for each individual at coding regions in *GST* genes relative to sites < 10 Mb upstream and downstream. Individuals with fewer than 500 sites scored within coding regions were omitted. **a** All regions. **b** Regions containing *GST* sweeps, comparing individuals with sweep backgrounds to those with no sweep backgrounds, and omitting populations with no sweep backgrounds. Boxplot centre lines indicate medians, hinges enclose the first and third quartiles, and whiskers extend to the largest value up to 1.5 times the interquartile range from each hinge. **a** Australia (*n* = 75); East Africa (*n* = 18); Indonesia (*n* = 36); Middle East (*n* = 17); North America (*n* = 18); Pacific Islands (*n* = 73); PNG (*n* = 57); South America (*n* = 17); South Asia (*n* = 17); Southeast Asia (*n* = 67). **b** Australia, no sweep/sweep (*n* = 31/11); Indonesia (*n* = 19/8); North America (*n* = 15/3); Pacific Islands (*n* = 27/6); PNG (*n* = 36/9); South America (*n* = 5/7); Southeast Asia (*n* = 9/4).

variants across populations using a latent factor mixed model as in *Ae. aegypti*, though restricted our analyses to the six populations with *VSSC* mutations. Sparse non-negative matrix factorisation run on 45,809 SNPs conditioned for genome-wide genetic structure, setting *K* = 4 (Supplementary Fig. 11). A second latent factor mixed model run on all populations identified no regions of strong association (Supplementary Fig. 12).

The model identified 19 SNPs around the *VSSC* gene that were strongly associated with F1534C (adjusted *P*-value < $2.183 \times 10^{-5}$) (Fig. 10b). A PCA of these SNPs indicated clustering among individuals with F1534C, and homozygotes in particular, suggesting a shared evolutionary history and a single substitution event underlying F1534C in these samples (Fig. 10c). Differences in Tajima's D (Fig. 10d) and π (Fig. 10e) between F1534C homozygotes and wild-type individuals from the same populations showed similar signs of positive selection as in *Ae. aegypti* (Fig. 5). Pairwise $F_{ST}$ between F1534C homozygotes and wild-type individuals likewise indicated a peak at the *VSSC* region (Fig. 10f).

Linkage network analysis was conducted with identical protocols to *Ae. aegypti* (section "Linkage disequilibrium networks in *Aedes*"

*aegypti*"), though as the *Ae. albopictus* genome assembly is far less contiguous we focused on the five contigs with the largest number of $r^2 > 0.6$ interactions. The five contigs all had peaks of strongly linked SNPs (Fig. 10g), though with considerably more noise possibly due to small sample size (*n* = 8). Some peaks overlapped genes with products linked to insecticide resistance, including cytochrome P450[4], glutathione S-transferase[1], acetylcholinesterase[6], UDP-glycosyltransferase[52], esterase B1[50], Nach sodium channel protein[3], xanthine dehydrogenase[57], and phenoloxidase-activating factor[58]. Genes identified through linkage network analysis are listed in Supplementary Data 10.

Several genes identified by linkage network analysis were common to both *Ae. aegypti* and *Ae. albopictus*. This included a cluster of cytochrome P450 genes of types including 6a8, 6a14, 6a20, 6d3, and 6d4. These genes are found on chromosome 2 in both species, as a cluster of 19 genes in *Ae. aegypti* and 16 genes in *Ae. albopictus*. This cluster was linked to two backgrounds in *Ae. aegypti* (V1016G and GST Americas) and the F1534C background in *Ae. albopictus* (Figs. 9, 10g). Both species also shared links to UDP-glucuronosyltransferase 2C1

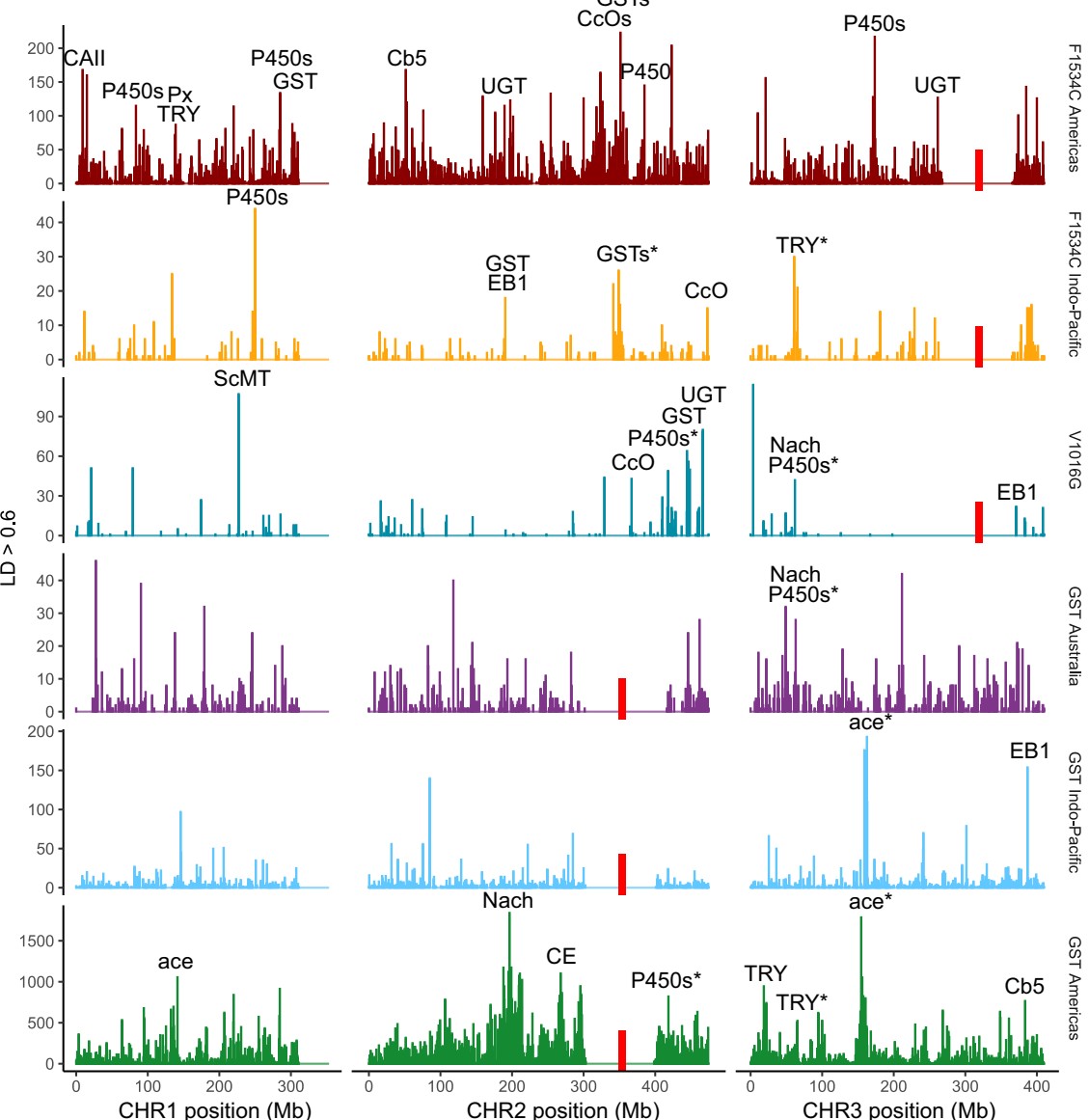

**Fig. 9 | Linkage network analysis in *Ae. aegypti*.** Rows indicate the three *VSSC* backgrounds (top) and the three *GST* backgrounds (bottom). Plots are histograms with 500 kb bins, showing locations of SNPs with r² > 0.6 to at least one SNP within 1 Mb of the sweep locus (red bars), and scoring SNPs for each r² > 0.6 interaction with a SNP near the locus. SNPs within 50 Mb of the sweep locus were omitted. Labels indicate peaks containing gene(s) of known resistance association: *P450* = cytochrome P450; *GST* = glutathione S-transferase; *ace* = acetylcholinesterase;

*UGT* = UDP-glycosyltransferase; *Nach* = sodium channel protein Nach; *CbS* = cytochrome b5; *CcO* = cytochrome c oxidase; *EB1* = esterase B1; *CE* = cholinesterase, *CAII* = carbonic anhydrase II; *Px* = peroxiredoxin-2; *TRY* = trypsin or anti-chymotrypsin; *ScMT* = sodium-coupled monocarboxylate transporter. Asterisks indicate peaks found in more than one background. Full list of genes in Supplementary Data 9, raw linkage data in Source Data.

genes found on chromosome 3 in *Ae. aegypti* and chromosome 2 in *Ae. albopictus*. These were linked to the F1534C Americas background in *Ae. aegypti*.

## Discussion

The spread of insecticide resistance in pests is a pressing global problem. Recent work in *Aedes* has shown how target-site resistance mutations at the *VSSC* gene can reach global distributions via gene flow following multiple independent substitutions[9,12,19]. Less is known about how metabolic resistance originates and spreads, though it is typically assumed to have more complex genetic architectures than target-site resistance[32]. This study identified three distinct genetic backgrounds at *GST* epsilon genes that have spread by positive selection across *Ae. aegypti* populations. The global scale and population genetic patterns

of these three genetic backgrounds are similar to the three *VSSC* backgrounds we also describe. While many different genes can contribute to metabolic resistance phenotypes, the observation that three evolutionarily independent backgrounds have undergone global partial sweeps at the same *GST* epsilon gene cluster strongly suggests this cluster has a major role in insecticide resistance. The global spread of *GST* backgrounds contrasts with those observed in anopheline mosquitoes where barriers to gene flow appear to have restricted the distribution of resistant alleles[59].

Although the *GST* and *VSSC* sweeps had similar broad geographical patterns, local patterns indicate these backgrounds spread mostly asynchronously. This is most evident in the Pacific Islands, where Vanuatu, Fiji, Kiribati, and New Caledonia all shared the same GST Americas haplotype, but each had only a single *VSSC* background

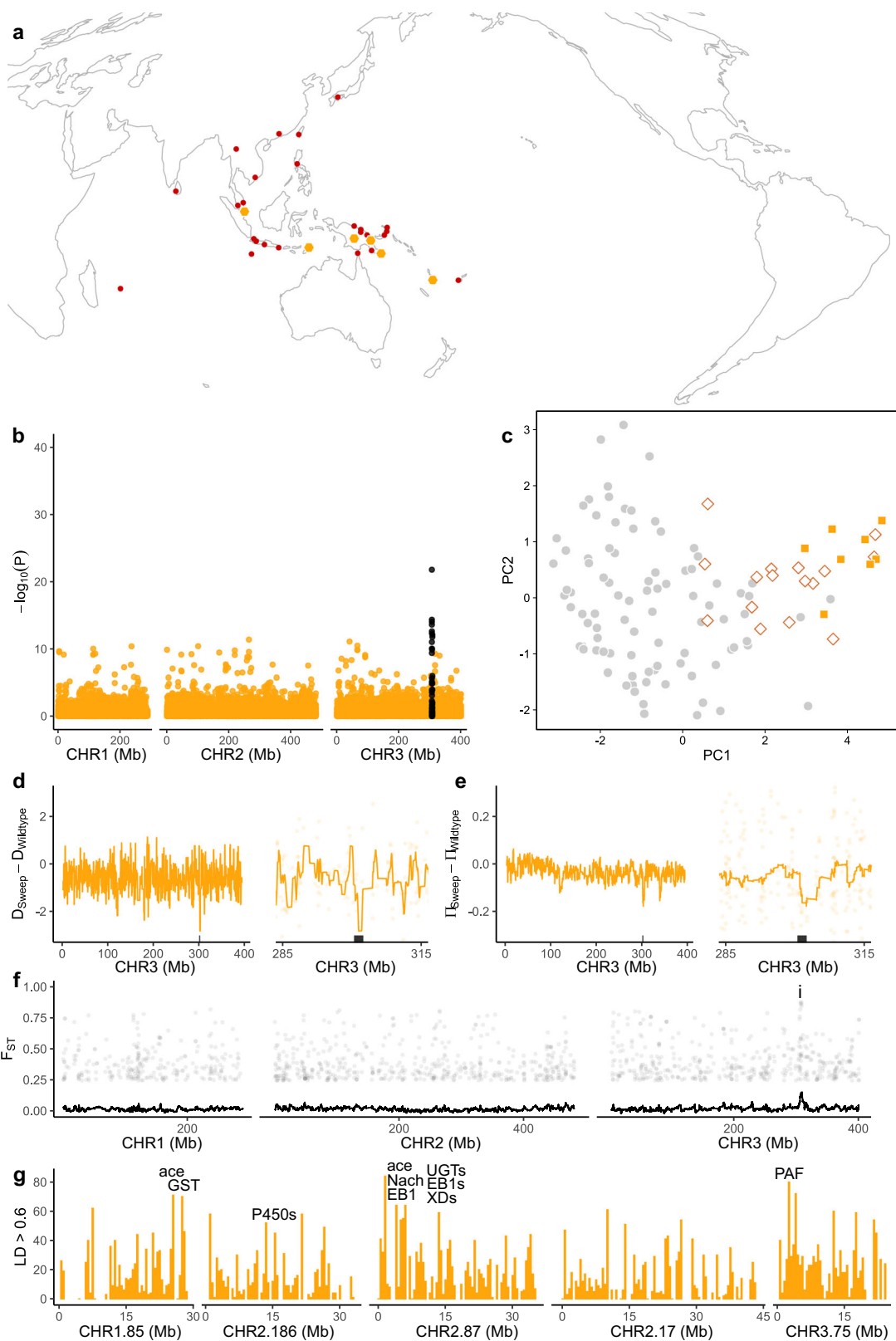

of either V1016G, F1534C Indo-Pacific, or F1534C Americas (Fig. 6). Vanuatu and Fiji were also at or near fixation for these *VSSC* backgrounds (Supplementary Data 3). However, this does not rule out the simultaneous spread of *GST* and *VSSC* backgrounds at some locations. Linkage network patterns add additional insight here, as both F1534C backgrounds were strongly linked to the *GST* epsilon region, but no *GST* backgrounds were linked to the *VSSC* gene (Fig. 9). Additional

temporal samples from key populations could help disentangle the shared evolutionary history of these genes.

The possible admixture patterns in New Caledonia provide additional insight into these invasion histories (Fig. 1). If these patterns reflect gene flow from an American background into an established Pacific Island background, this could mark the point at which the GST Americas haplotype was brought into the Pacific Islands. As F1534C is

**Fig. 10 | Genomic characterisation of the F1534C mutation at the voltage-sensitive sodium channel (*VSSC*) gene in *Ae. albopictus*. a** Orange hexagons indicate populations with F1534C, red circles indicate other populations. Local frequencies of each mutation are listed in Supplementary Data 4. **b** Latent factor mixed model identifying genome-wide SNPs with genetic structure in line with *VSSC* genotype after controlling for genome-wide patterns assessed using sparse non-negative matrix factorisation with $K = 4$ (Supplementary Fig. 11). Black circles indicate SNPs within 1 Mb of the *VSSC* gene on chromosome 3. Only the six populations with F1534C were included in the model. *P*-values were adjusted from combined z-scores using a Benjamini-Hochberg correction with a false discovery rate of 0.01. **c** PCA of 19 SNPs strongly associated with F1534C. Orange squares = F1534C homozygote, white filled lozenges = heterozygote, grey circles = wild type. **d** Difference in Tajima's D between sweep-associated individuals and wild-type individuals from the same six populations as the sweeps. The left-hand plot shows all of chromosome 3, and the right-hand plot focuses on the gene region. Black

rectangles indicate the *VSSC* gene region ± 1 Mbp. Coloured lines show moving averages. **e** As above but showing differences in nucleotide diversity. **f** Pairwise $F_{ST}$ between individuals of the swept background and wild-type individuals from the same six populations. SNPs with $F_{ST} > 0.25$ are plotted. The *VSSC* region is seen as a peak on chromosome 3, indicated by 'i'. **g** Linkage network analysis. Plots are histograms with 500 kb bins, showing locations of SNPs with $r^2 > 0.6$ to at least one SNP within 1 Mb of the sweep locus, and scoring SNPs for each $r^2 > 0.6$ interaction with a SNP near the locus. SNPs within 50 Mb of the sweep locus were omitted. The five most strongly associated contigs are plotted. Labels indicate peaks containing genes of known resistance association: *P450* = cytochrome *P450*; *GST* = glutathione S-transferase; ace = acetylcholinesterase; EB1 = esterase B1; *UGT* = UDP-glycosyl-transferase; *Nach* = sodium channel protein Nach; *XD* = xanthine dehydrogenase; *PAF* = phenoloxidase-activating factor 2. Full list of genes in Supplementary Data 10, raw linkage data in Source Data.

at a moderate frequency in New Caledonia ($p = 0.21$), it is conceivable that the GST Americas haplotype spread from New Caledonia into Fiji, Kiribati, and Vanuatu by *VSSC* wild-type mosquitoes, though whether this occurred before or after the local introduction of *VSSC* mutations is unclear. New Caledonia was a hub of international shipping from the late 19th century to WWII[60]; given that *Ae. aegypti* rapidly reaches high local densities after colonisation, a large number of mosquitoes must have been introduced from the American source to produce genome-wide admixture patterns as strong as those observed[15]. If admixture occurred before the 1960s, this likely predates the introduction of the GST Americas haplotype, as insecticidal control efforts only began in New Caledonia in the 1960s[60]. There is also some confusion around the source and timing of the *Ae. aegypti* invasion into the Pacific Islands. While most evidence points to an invasion from the Mediterranean at the end of the 19th century[21], an earlier invasion of the Pacific Islands from the Americas is plausible, given historical records of arboviral disease outbreaks in Southeast Asia[61] and early outbreaks of dengue in New Caledonia specifically[60]. The PCA results and non-significant $F_3$ tests suggest that if this hypothesis is true, it may be too old or has experienced too much local gene flow to be detected[37].

Sweeps at the *VSSC* gene align with previous evidence that the F1534C allele is derived from two identical substitutions[12,19] and that both F1534C and V1016G have swept through the Indo-Pacific[9]. Here, we detail the extent to which each *VSSC* background has spread, most notably the F1534C Americas background, which was found not only in the Americas but at locations in the Pacific Islands, Southeast Asia, and the Middle East (Fig. 4c). Interestingly, while no *VSSC* wildtypes were observed in Singapore, Malaysia, Saudi Arabia, Taiwan, or PNG, these countries contained both F1534C and V1016G at intermediate frequencies (Supplementary Data 3), and these may stay as partial sweeps if neither confers a fitness advantage sufficiently large to replace the other. A recent study identified another *VSSC* mutation, L982W, that produced strong resistance phenotypes when combined with F1534C but has not been recorded outside of Vietnam and Cambodia[62]. Here, F1534C was observed in *Ae. aegypti* samples from Ho Chi Minh City and Nha Trang, but only as a heterozygote (Supplementary Data 1). If the L982W mutation is presently spreading through any other Indo-Pacific populations, it may be detectable as a third F1534C background, particularly if this background spreads through populations that are already at or near fixation for *VSSC* mutations. This assumes the new haplotype confers an additional selective advantage in wild populations.

Compared with *Ae. aegypti*, *Ae. albopictus* had far fewer populations with *VSSC* resistance alleles (Fig. 10a; c.f. 4c), and when present these alleles were at lower frequencies (Supplementary Data 3, 4). Despite the comparative rarity of these alleles, a single F1534C allele copy was found in the most geographically remote population, in North Fly Region, PNG. F1534C was found at low-frequency in three PNG populations >600 km apart and may be present in other

unsampled locations such as in the South Fly Region, which has a similar genetic background to Torres Strait Islands *Ae. albopictus* and is the likely source of admixture observed in North Fly Region[35]. The spread of this allele across six populations points to a selective advantage, and low frequencies of this allele compared with those in *Ae. aegypti* might be due to limited insecticide exposure in sylvan habitats where *Ae. albopictus* is more common than *Ae. aegypti*[63]. The current geographical distribution of F1534C and its apparent selective advantage suggest future insecticide usage against *Ae. albopictus* may lead to F1534C rapidly increasing in frequency.

There was substantial variation in *GST* epsilon class read depth among regions, with North America ~8.5× higher than the Middle East and South Asia (Fig. 8). This copy number increase likely occurred after the GST Americas sweep, as the GST Americas haplotype was rare in New Mexico while the copy number increase was observed in almost all individuals. This inference aligns with the observation that none of the *GST* sweeps were linked to gene duplications (Fig. 8b), and together these point to different mutational mechanisms underpinning the sweeps, such as point mutations; a single mutation in a *GST* epsilon gene was linked to metabolic resistance to DDT in Beninese *An. funestus*[30]. The New Mexico population in this study is part of an *Ae. aegypti* genetic cluster extending from Southern USA through to Central America and the Caribbean[64], and it would be valuable to explore copy number variation and associated resistance further in this region. Positive selection was recently detected at this *GST* gene cluster in Brazil and Colombia[65] which likely reflects the GST Americas background in these populations.

The GST Australia sweep likewise demands further investigation. Usage of insecticides against mosquitoes in Australia has been low since DDT was banned in 1987[66]. The GST Australia sweep could reflect resistance evolving and spreading in response to indirect insecticide use such as in agriculture, or may represent a second selective sweep at the *GST* locus of a wild-type background after an initial resistant sweep; both of these phenomena have been observed in *Culex* mosquitoes[13]. If the sweep represents a wild-type background this also implies a considerable fitness cost of the resistant allele like that reported for the F1534C mutation in *Ae. aegypti*[24]. The partial nature of the *GST* sweeps is consistent with fitness costs, with no evidence of fixation in any of the populations containing sweeping haplotypes. Future work should aim to understand the fitness benefits and costs of the three *GST* sweeps, including whether these confer resistance to pyrethroids or other insecticides (e.g., organophosphates or carbamates) and how this resistance interacts with *VSSC* mutations. The three *GST* sweeps may also vary in which insecticides they resist, as seen in F1534C and V1016G[3].

Although this study covers the geography of the Indo-Pacific region extensively, our investigation of *Ae. aegypti* had only single samples from each of Africa, North America, and South America. This limited our capacity to infer admixture sources and to investigate the

geographical coverage of the increased *GST* copy number in New Mexico. Furthermore, while our ddRAD data provided strong evidence of sweeps and copy number variation, the sparseness of ddRAD data prevented us from inferring specific *GST* genes that had undergone duplication. This could be further investigated with whole genome data or specific sequencing of this region. Our analysis of *Ae. albopictus* was also limited by a lack of samples from outside the Indo-Pacific region, as well as the incompleteness of this species' genome assembly relative to that of *Ae. aegypti*. A broader sampling of *Ae. albopictus* may also provide material for characterising the V1016G mutation which has been recorded recently in this species[23] but was not detected in any of our samples.

In summary, our analysis of *Ae. aegypti* and *Ae. albopictus* has uncovered a set of evolutionarily independent, globally segregating, partial selective sweeps at multiple resistance genes that have asynchronously spread resistance alleles across geographically distant populations. The complex patterns uncovered in these invasive disease vectors provide a detailed picture of insecticide-based selection worldwide that can help guide chemical control options for local authorities[2,32]. These results highlight the *GST* epsilon class genes as having a major global impact. Our findings also indicate a set of specific resistance backgrounds to be considered in novel mosquito control strategies that rely on matched resistance levels between lab and field populations for long-term success[67].

## Methods

Our study did not require ethical approvals or sampling permits, except for the *Ae. aegypti* from New Mexico which were imported as live eggs under the Department of Agriculture and Water Resources Permit for conditionally non-prohibited goods (Permit No. 0002631825) and Department of the Environment and Energy NON-CITES permit PWS2019-AU-001275. These samples were reared in the laboratory before processing. All other samples were collected from the field as either adults or immatures. Sampling of adult mosquitoes was either by sweep-netting or BG-Sentinel mosquito traps (Biogents AG, Regensburg, Germany). Sampling of immature stages was by ovi-trap or pipetting from larval habitats.

### Study design

This study used genome-wide sequence data and endpoint genotyping assays to identify genetic backgrounds at insecticide resistance genes that showed signs of positive selection and were found across genomically distinct populations of *Aedes* mosquitoes. We used genetic data from 26 countries. This includes 32 populations of *Ae. aegypti* from 19 countries (*n* = 444) and 40 populations of *Ae. albopictus* from 17 countries (*n* = 490). Samples specifically sequenced for this project included *Ae. aegypti* (*n* = 148) from 12 populations (Kenya, Tonga, Timor-Leste, USA (New Mexico), Australia (Cape York and Torres Strait Islands) and PNG (East New Britain, East Sepik, Lae, Madang, Milne Bay, and Port Moresby)), and *Ae. albopictus* (*n* = 210) from 12 populations (Indonesia (Yogyakarta) and PNG (Alotau Town, Duke of York Island, East New Britain, Kokopo, Lae, Lihir Island, Malba, Mawi, North Fly Region, Vanimo, Wambisa, Wewak). South Fly Region in PNG could not be sampled due to COVID-19 restrictions on fieldwork. The *Ae. aegypti* sample from New Mexico was a lab strain in its fourth generation[68]. All samples, including those used from previous projects, are listed in Supplementary Data 1, 2. Genomic DNA was extracted from the 148 *Ae. aegypti* and 210 *Ae. albopictus* using Qiagen DNeasy Blood & Tissue Kits (Qiagen, Hilden, Germany) or Roche High Pure™ PCR Template Preparation Kits (Roche Molecular Systems, Inc., Pleasanton, CA, USA).

### qPCR assays for *VSSC* mutations

*Aedes aegypti* were screened for F1534C, V1016G, and S989P mutations using endpoint genotyping assays (Custom TaqMan™ SNP assays, Thermo Fisher Scientific, Waltham MA, USA, Cat. No. 4332077)[9].

Genotyping assays were run on a LightCycler II 480 (Roche, Basel, Switzerland) real-time PCR machine in a 384-well format. The PCR Master Mix contained 40× Custom TaqMan™ assay (0.174 µL), 2 × KAPA™ Fast PCR Probe Force qPCR Master Mix (3.5 µL) (Kapa Biosystems, Wilmington MA, USA, Cat. No. 07959338001), ddH$_2$O (1.326 µL) and genomic DNA (2 µL). PCR conditions comprised a pre-incubation step of 3 min at 98 °C (ramp rate 4.8 °C/s) followed by 40 cycles of amplification at 95 °C for 10 s (2.5 °C/s ramp rate) and 60 °C for 20 s (2.5 °C/s ramp rate; Acquisition mode: single) with a final cooling step of 37 °C for 1 min (2.5 °C/s ramp rate).

*Aedes albopictus* were prepared for Sanger sequencing of domain III of the *VSSC* gene to characterise DNA sequences at codon 1534, using primers aegSCF7 (GAGAACTCGCCGATGAACTT) and aegSCR7 (GACGACGAAATCGAACAGGT)[69] to generate an amplicon of 740 bp. A PCR master mix was employed with final concentrations of Standard ThermoPol buffer Mg-free (1x) (New England Biolabs, Ipswich MA, USA), dNTPs (0.2 mM each) (Bioline, London UK), MgCl2 (1.5 mM) (Bioline, London UK), 0.5 µM each of forward and reverse primers, 0.625 units of Immolase™ Taq polymerase (Bioline, London, UK), 2 µL genomic DNA and PCR-grade H2O, to a final volume of 25 µL. PCR cycling conditions were: initial denaturation of 95 °C for 10 min, 35 cycles of 95 °C for 30 s, annealing at 52 °C for 45 s and extension at 72 °C for 45 s, followed by a final extension of 5 min at 72 °C. Amplicons were purified and sequenced by Macrogen Inc. in Seoul, Korea, on a 3730xl DNA analyser using sequencing primers Alb171F (CCGATT CGCGAGACCAACAT)[70] and aegSCR8 (TAGCTTTCAGCGGCTTCTTC)[69]. Sequences were analysed using Geneious® 11.1.4 (Biomatters Ltd).

### Sequencing and processing genomic data

Extracted DNA was used to build double digest restriction-site associated DNA (ddRAD[71]) sequencing libraries following the protocol of Rašić, Filipović, Weeks, & Hoffmann[72]. Libraries were individually barcoded and sequenced on either a HiSeq 4000 or a Novaseq 6000 using 150 bp chemistry. New and old sequence data were combined for each species and run through the same bioinformatic pipeline. We used the Stacks v2.54[73] programme "process_radtags" to demultiplex sequences and remove sequences with Phred scores below 20. Sequences were aligned to the nuclear genome assembly for *Ae. aegypti*, AaegL5[74], and the linkage-based assembly for *Ae. albopictus*, AalbF3[75], using Bowtie2 v2.3.4.3[76] with "−very-sensitive" settings. The output .bam files were used for genomic data analysis using either Stacks (sections "Genetic structure and admixture", "Genome-wide association with *VSSC* mutations" and "Genome-wide analysis of introgressed backgrounds") or GATK v4.2.6.1[77] (section "Copy number variation at glutathione S-transferase (*GST*) genes").

### Genomic data analysis

**Genetic structure and admixture.** Sequences were built into Stacks catalogues for each species using the Stacks programme "ref_map". The Stacks programme "populations" was used to export VCF files containing SNP genotypes for all individuals in each catalogue, filtering to retain SNPs called in at least 50% of individuals from each population and 90% of individuals total, and with a minor allele count ≥ 3[78]. To avoid issues from low sample numbers, for the above filtering steps, we treated PNG *Ae. aegypti* samples from East Sepik and Lae as one population, and Torres Strait Islands *Ae. albopictus* as one population (Supplementary Datas 1, 2). All samples had < 30% missing data, and the total missing data was low (Ae. aegypti, $\bar{x}$ = 2.42%; Ae. albopictus, $\bar{x}$ = 4.88%). For analyses of genetic structure and latent factor mixed models, missing data were imputed in windows of 6000 SNPs using Beagle v4.1[79], using 10 iterations, 500 SNP overlaps, and the full dataset as reference. The final datasets contained 79,084 SNPs for *Ae. aegypti* and 96,269 SNPs for *Ae. albopictus*.

Genome-wide genetic structure among individuals was analysed using fineRADstructure[33]. Relevant subsets of populations were further

analysed with TreeMix v1.13[36] to build maximum likelihood trees and run $F_3$ tests[80]. TreeMix analysed SNPs in blocks of 1000, using a bootstrap replicate, and with the root set to Kenya (*Ae. aegypti*) or China (*Ae. albopictus*). Principal components analysis was run in R package SNPRelate v1.34.1[81], using functions "snpgdsPCA" to build a principal component axis from each pair of source populations, "snpgdsPCASNPLoading" to calculate SNP loadings, and "snpgdsPCA-SampLoading" to project individuals onto the principal component axis.

**Genome-wide association with *VSSC* mutations.** Latent factor mixed models associating SNP variation with *VSSC* genotype were run in the R package LEA v3.2.0[82]. Analysis of *Ae. aegypti* used all samples with available genomic and *VSSC* data ($n = 395$). Analysis of *Ae. albopictus* used all samples from the six populations where *VSSC* mutations were found ($n = 114$); analysis of all samples ($n = 490$) found no associations (Supplementary Fig. 12), potentially due to the much lower frequency of *VSSC* mutations in *Ae. albopictus*.

As both species exhibited strong genome-wide genetic structure among populations, we first used sparse non-negative matrix factor-isation (function "snmf") to determine an appropriate number of K clusters with which to condition the mixed models. Sparse non-negative matrix factorisation used 10 repetitions for each potential K, which identified $K = 18$ for *Ae. aegypti* (Supplementary Fig. 1) and $K = 4$ for *Ae. albopictus* (Supplementary Fig. 11). Latent factor mixed models (function "lfmm") were run using these K and a minor allele frequency of 0.05, and with 10,000 iterations, a burning of 5000, and 10 repetitions.

Adjusted *P*-values were computed from the combined z-scores, using a Benjamini-Hochberg correction with a false discovery rate of 0.01. Strongly associated SNPs were those that had adjusted *P*-values below the inverse of the number of SNPs ($1.956 \times 10^{-5}$ for *Ae. aegypti*; $2.183 \times 10^{-5}$ for *Ae. albopictus*). Strongly associated SNPs were found in clusters around the *VSSC* and *GST* regions; these were used in PCAs (Figs. 4b, 10c) and heatmaps (Fig. 6b). Principal components analyses were run in LEA (function "pca").

**Genome-wide analysis of introgressed backgrounds.** Tajima's D, nucleotide diversity, linkage disequilibrium, and pairwise $F_{ST}$ were calculated in VCFtools v0.1.16 using the filtered dataset without the imputation step. Tajima's D was calculated in bins of 5 kb using "--TajimaD 5000". Nucleotide diversity was calculated using "--site-pi". Linkage disequilibrium was calculated as the squared correlation coefficient between genotypes ($r^2$) using "--geno-r2", setting "--ld-window-bp-min 500" and "--ld-window-bp 100000" to ignore compar-isons within RADtags or separated by more than 100 kb, and this was visualised using the average chromosome position and $r^2$ scores of each bin. $F_{ST}$ was calculated using "--weir-fst-pop", "--fst-window-size 5000" and "--fst-window-step 1000". Parameters were visualised as moving averages using the R package "tidyquant" (function "geom_ma"; ma_fun = SMA).

Linkage network analysis used VCFtools to calculate intrachro-mosomal ("--geno-r2") and interchromosomal ("--interchrom-geno-r2") linkage disequilibrium for the six subsets of individuals associated with each sweep, plus the three subsets of wild-type individuals from the same populations as the *GST* sweeps, plus the total set of *VSSC* wild types. Results were visualised with the R package "tidyverse" (function "geom_histogram"; binwidth = 1). The $r^2 > 0.6$ cutoff was chosen fol-lowing iterative evaluation of cutoffs from 0.3 to 0.8, with $r^2 > 0.6$ providing the strongest visual signal-to-noise ratio.

**Copy number variation at glutathione S-transferase (*GST*) genes.** Copy number variation was assessed by comparing read depths at *GST* genes relative to those upstream and downstream. This required a genotyping and filtering pipeline that retained monomorphic as well as polymorphic sites. First, .bam files were processed in samtools v1.16 to remove unmapped reads and non-primary alignments. GVCF files were produced for each individual using HaplotypeCaller in GATK v4.2.6.1[77], and these were genotyped using GenotypeGVCFs set to '--include-non-variant-sites'. Hard filtering excluded indels then followed standard GATK guidelines for non-model taxa, setting "QD < 2.0", "QUAL < 30.0", "SOR > 3.0", "FS > 60.0", and "MQ < 40.0". Individuals were then filtered individually in bcftools v.1.16[83] to remove missing data sites, sites with star alleles, and sites with less than 1X read depth. Read depths were calculated in vcftools (function "--depth"), using all sites retained after filtering. Depths at *GST* genes were calculated as the average depth across the 15 *GST* coding regions. Individuals with fewer than 500 genotyped sites within the 15 coding regions were omitted.

Copy number variation was calculated at the *Ae. aegypti VSSC* gene using the same methods as above, with read depth ratios calcu-lated from the *VSSC* coding region.

### Geographical mapping
All maps were produced using ArcGIS Pro v3.1 (https://www.esri.com/en-us/arcgis/products/arcgis-pro/overview). World basemaps were derived from land and ocean shapefiles in the public domain available from Natural Earth (https://www.naturalearthdata.com). Basemaps used a Mollweide projection with a central meridian of 160° E.

### Reporting summary
Further information on research design is available in the Nature Portfolio Reporting Summary linked to this article.

### Data availability
Raw .fq files and relevant metadata for 934 mosquitoes is available at the NCBI SRA at https://www.ncbi.nlm.nih.gov/bioproject/?term=PRJNA1117085 (*Aedes aegypti*) and https://www.ncbi.nlm.nih.gov/bioproject/?term=PRJNA1118433 (*Aedes albopictus*). List of samples used in genomic analyses are included in Supplementary Data 1 and Supplementary Data 2. Source data are provided with this paper.

### Code availability
Code used in processing, analysis, and plotting is provided with this paper as Supplementary Code 1.

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

## Acknowledgements

We acknowledge the efforts of Jiannong Xu, Esther Anderson, Scott Ritchie, and Siaola Mahe for mosquito collections and shipments. We thank Frank Jiggins, Gabriela Montejo-Kovacevich, and Henry North for useful early discussions on analysis. We acknowledge the efforts of the PNGIMR entomology team, especially Samuel Demok, Peter Kaman, and Joelyn Goi, as well as the PNGIMR LF team for their assistance with mosquito collections and shipments. We acknowledge the support of Tim Freeman and RAM team for work conducted in National Capital District, Papua New Guinea. We acknowledge the support of Leo Makita for samples collected during malaria control surveys. We acknowledge PNG Provincial Health Authorities in East New Britain, New Ireland, East Sepik, West Sepik, Morobe, Madang, Milne Bay, Western and National Capital District for their collaboration and support. We acknowledge the efforts of the STRIVE PNG Partnership Project Managers Annie Dori and Rachael Farquhar for the coordination they facilitate across multiple organisations to support the implementation of activities under the leadership of the PNG National Malaria Control Programme. Sampling and sequencing of Papua New Guinea was funded by STRIVE PNG through the Department of Foreign Affairs and Trade, Australian Government [Grant No. 74430], Indo-Pacific Centre for Health Security. T.L.S. was funded by an ARC DECRA Fellowship (DE230100257). L.J.R. was funded by a NHMRC Career Development Fellowship (GNT1161627).

SK was funded by NHMRC Development Grant (GNT1141441) and NHMRC Ideas Grant (2004390).

## Author contributions

A.A.H., S.K., L.J.R., and M.L. acquired funding; T.L.S., N.M.E.H., A.A.H., and S.K. designed the project; A.A.H., S.K., L.J.R., and M.L. provided resources; M.K., R.V., M.L., and S.K. conducted fieldwork; A.A.H., S.K., and M.L. provided supervision; A.A.H., S.K., L.J.R., M.L. and N.M.E.H. provided project administration; N.M.E. processed samples with help from T.L.S., and A.R.V.R. N.M.E.H. performed endpoint genotyping assays with help from ARJvR; NMEH generated ddRAD libraries with help from T.L.S.and A.R.V.R. T.L.S., N.M.E.H., A.R.V.R., S.K., M.K., and R.V. curated the data; T.L.S. designed and performed genomic analyses; T.L.S. developed visualisations; T.L.S. and N.M.E.H. validated the results; T.L.S. wrote the paper with help from N.M.E.H. T.L.S., N.M.E.H., A.A.H., L.J.R. and S.K. edited the paper.

## Competing interests

The authors declare no competing interests.
