## [Peer Review File · Nature Communications]

Global, asynchronous partial sweeps at multiple insecticide resistance genes in *Aedes* mosquitoesReviewers' Comments:

Reviewer #1:

Remarks to the Author:

This manuscript describes work on a large dataset comprising of RADseq and focal genotyping results from *Aedes aegypti* and *Ae. albopictus* primarily from multiple countries in Asia and Australasia, with additional samples from Africa, the middle eastern region and the Americas. The work focuses initially on patterns of genetic structure of the samples in each species using haplotype similarity analysis. Patterns are broadly consistent with geography or source of recent invasive populations, although some admixed populations are identified and these are investigated further to identify sources of mixture. Following identification of patterns of genetic structure using clustering analysis, outlier SNPs were identified which segregated with Vssc V1016G and F1534C genotypes rather than genetic structure. The majority were around the Vssc gene on chromosome 3 but interestingly a smaller number were located in the epsilon GST cluster on chromosome 2. Outlier Vssc SNPs clustered separately according to background (1534C Americas; 1534C Indo-Pacific and 1016G) with each Vssc group showing signals of selective sweeps relative to wild type and also strong differentiation among them indicative of different evolutionary origins. In addition several other regions containing known or putative resistance genes showed strong differentiation between American and Indo-Pacific backgrounds. Three geographical GST groups were identified and each showed signals of selective sweeps similar to the Vssc analysis previously, and strong differentiation between groups indicating different origins. The Mexican sample but not others also showed high read depth ratios at the GST genes demonstrating copy number variation. In contrast to *Ae aegypti*, *Ae albopictus* showed only a single origin. Overall the results indicate global but distinct spread of both Vssc (target site) and GST (metabolic) resistance variation.

The manuscript is generally well written with consequential results for understanding the evolution of insecticide resistance in both species and particular novelty in identification of the sweeps and by inference global importance of the GST genes alongside the well known Vssc mutants.

Line 109 What is meant by 'putative wild type samples'

Line 146 How were the subsets of populations for treemix chosen – based on patterns from Figs 1a, 2a or some other criteria?

Line 158 It would be helpful to clarify why the link to the branch linking to Kenya indicates an American background.

Scale for Figs 1,2 missing from legend and consider changing colour scheme to make clearer

Line 159 for clarity add 'other' before PNG

Line 169 The interpretation as to why F3 results were significant for *Ae albopictus* but not *Ae aegypti* does not appear clear and merits some discussion and maybe inclusion of a figure

Line 218 onwards it would be clearer to refer to individuals homozygous for the mutation as 1016G (or 1534C as appropriate)

Line 262-263 this is a very large region overlapping the eGST genes – how big is the eGST region itself and is it central to this? From Fig 6a it seems potentially so for the 1534C associated SNPs but much less so for the 1016G associated SNPs. What other genes are in this region? It might be helpful to report the SNPs in a supplementary table.

Reviewer #2:

Remarks to the Author:

Dear Authors,

I recently had the opportunity to review your manuscript. I commend you on the comprehensive analysis and clear presentation of your research on insecticide resistance in *Aedes aegypti* and *Ae. albopictus*. The depth of your study is impressive, and it provides significant insights into the genetic

mechanisms of resistance in these crucial vector species.

I do, however, have a few queries and suggestions that I believe could further enhance the impact and utility of your work:

Imputation Methodology: Your decision to use Beagle for imputation piqued my interest. Could you elaborate on why you chose this method? Specifically, I'd like to know whether you utilized micro haplotypes for data input or relied solely on statistical imputation.

Linkage Network Analysis: Have you considered employing linkage network analysis, perhaps using LDna, to examine the genetic architecture in greater detail? Such an analysis could provide additional layers of understanding regarding the genetic relationships and dynamics within and between the studied populations.

Code Sharing and Reproducibility: I am thoroughly impressed with your analytical approach. To further benefit the scientific community, especially graduate students and researchers looking to replicate or build upon your work, share the detailed code of your analysis. Creating a step-by-step guide in a format like Quarto or Markdown, accompanied by all necessary files and raw data, would be immensely valuable. Platforms like Zenodo or Dryad are excellent for depositing these materials. While I understand this is at your discretion, it would significantly contribute to the study's transparency and facilitate academic collaboration.

In my review of your manuscript, I have included several suggestions directly in the Word document for your consideration. Please note that some of these suggestions reflect American English conventions, and you are welcome to disregard them if they do not align with your preferred style or dialect. I intend to enhance the clarity and readability of your manuscript while fully respecting your linguistic preferences and writing style.

Your work is undoubtedly a significant contribution to the field, and these suggestions aim to maximize its impact and accessibility. Thank you for the opportunity to review your manuscript. I look forward to its continued development and contribution to our understanding of mosquito-borne disease control.

Reviewer #3:

Remarks to the Author:

Historically, GST is well known to be involved in DDT, organophosphate, and carbamate insecticide resistance. Some papers have reported the association between GST and pyrethroid resistance but to my best knowledge, no clear conclusion has been reached. Historically, both pyrethroids and organophosphates have been used to control *Aedes* mosquitoes and it is not surprising that genes associated with resistance to these two groups of insecticides are accumulating in mosquitoes. Most experts whose major is insecticide resistance will simply understand that two genes (regions) have been selected by use of organophosphates and pyrethroids. I think the authors should describe the importance of the paper more clearly.

Unfortunately, in this paper, no evidence whether any of the 15 GSTs found in this study are involved in resistance to any insecticide are indicated. If the authors show the clear evidence that any of the GSTs confer pyrethroid resistance, I believe it is worth to be published in this journal. But without showing such evidential data, I believe this paper fails to present results that exceed the expectations of many insecticide resistance researchers.

The knockdown resistance mutations found in the voltage sensitive sodium channel (Vssc), the target site of pyrethroid insecticide, is well studied in *Aedes aegypti*. It is well known that V1016G and F1534C are widely distributed in the world. These are the major mutations in *Ae. aegypti* but some

other mutations, like V253F and V410L, have also proven to be involved in pyrethroid resistance. In *Ae. albopictus*, F1534C is the major mutation that confers pyrethroid resistance but recently, V1016G has also been detected in several countries including Italy, Vietnam, China, Bulgaria, France, Georgia, Malta, Romania, Spain, Switzerland, and Turkey. So the authors should show the genetic data of mosquitoes having these other kdr mutations.

Line 356: This is wrong. There has been no haplotype having both L982W and V1016G.

Generally, purpose of the study is not clear. The last three lines of the summary should be more specific as to what is written, i.e., how clarification of the resistance background is useful for vector control.

I think that the entire paper needs to be written in a more reader-friendly manner. For instance, the word "sweep" or "swept" is difficult to understand for general readers of the journal.

REVIEWER COMMENTS

Reviewer #1 (Remarks to the Author):

This manuscript describes work on a large dataset comprising of RADseq and focal genotyping results from *Aedes aegypti* and *Ae. albopictus* primarily from multiple countries in Asia and Australasia, with additional samples from Africa, the middle eastern region and the Americas. The work focuses initially on patterns of genetic structure of the samples in each species using haplotype similarity analysis. Patterns are broadly consistent with geography or source of recent invasive populations, although some admixed populations are identified and these are investigated further to identify sources of mixture. Following identification of patterns of genetic structure using clustering analysis, outlier SNPs were identified which segregated with Vssc V1016G and F1534C genotypes rather than genetic structure. The majority were around the Vssc gene on chromosome 3 but interestingly a smaller number were located in the epsilon GST cluster on chromosome 2. Outlier Vssc SNPs clustered separately according to background (1534C Americas; 1534C Indo-Pacific and 1016G) with each Vssc group showing signals of selective sweeps relative to wild type and also strong differentiation among them indicative of different evolutionary origins. In addition several other regions containing known or putative resistance genes showed strong differentiation between American and Indo-Pacific backgrounds. Three geographical GST groups were identified and each showed signals of selective sweeps similar to the Vssc analysis previously, and strong differentiation between groups indicating different origins. The Mexican sample but not others also showed high read depth ratios at the GST genes demonstrating copy number variation. In contrast to *Ae aegypti*, *Ae albopictus* showed only a single origin. Overall the results indicate global but distinct spread of both Vssc (target site) and GST (metabolic) resistance variation.

The manuscript is generally well written with consequential results for understanding the evolution of insecticide resistance in both species and particular novelty in identification of the sweeps and by inference global importance of the GST genes alongside the well known Vssc mutants.

>>> We thank the Reviewer for their helpful summary of the MS and their insightful comments. We share the reviewer's views on our insecticide resistance findings as being consequential and novel.

Line 109 What is meant by 'putative wild type samples'

>>> These referred to samples which did not have the GST sweep haplotype.

We have changed these sentences to read: "... our analysis also detected another genomic region in *Ae. aegypti* which has undergone three evolutionarily independent, partial selective sweeps, each segregating across multiple populations. Comparisons with other individuals from these populations showed ..."

Line 146 How were the subsets of populations for treemix chosen – based on patterns from Figs 1a, 2a or some other criteria?

>>> In each case, TreeMix runs included the putative admixed populations identified with fineRADstructure (New Caledonia in 1a | North Fly in 2a), plus all populations from the putative source clades of admixture (Americas and West Pacific | PNG and Indonesia), plus an outgroup (Kenya | China). For *Aedes aegypti*, we also included Australian populations as these formed a clade between the American and West Pacific populations.

We have clarified this in the main text. “We ran TreeMix on subsets of populations selected from the fineRADstructure analysis, including putatively admixed populations, populations from potential source clades, and populations from clades intermediate to them.”

Line 158 It would be helpful to clarify why the link to the branch linking to Kenya indicates an American background.

>>> We have rewritten this to allow for the possibility of African introgression, though this is less likely given the haplotype clustering in Fig 1a.

“Here, *Ae. aegypti* from New Caledonia appeared to be an admixture of an American or African background (most likely American given Fig 1a) ...”

Scale for Figs 1,2 missing from legend and consider changing colour scheme to make clearer

>>> We have changed the colour scheme of the fineRADstructure plots, so that all ‘minimum coancestry’ values are now coded as white rather than bright yellow. We have also changed the colour gradient. This has greatly enhanced the discernability of higher coancestry pairs. We have added scales for the fineRADstructure and TreeMix subfigures.

Line 159 for clarity add ‘other’ before PNG

>>> We have added this.

Line 169 The interpretation as to why F3 results were significant for *Ae albopictus* but not *Ae aegypti* does not appear clear and merits some discussion and maybe inclusion of a figure

>>> We have added the following text:

“No F3 tests were significant for any *Ae. aegypti* populations, including comparisons involving New Caledonia. This result may reflect limitations of F3 tests in detecting older or asymmetrical admixture ³⁶, which may be the case in New Caledonia, and may also reflect our limited sampling of American *Ae. aegypti* populations.”

Overall, there are several confounding factors that prevent us understanding the evolutionary history of *Aedes aegypti* in New Caledonia. Whatever limitations there are of F3 tests, the limited access to American populations may be the key driver of this null result. Of the two included, New Mexico and Rio de Janeiro, neither is likely the true source, in that New Mexico is landlocked and the Rio de Janeiro *Aedes aegypti* population of pre-WWII has been likely replaced following local eradication efforts in the latter half of the 20th century. American Pacific populations such as Hawaii could be the true source.

Line 218 onwards it would be clearer to refer to individuals homozygous for the mutation as 1016G (or 1534C as appropriate)

>>> We have changed the language to describe individuals with VSSC mutations throughout.

Line 262-263 this is a very large region overlapping the eGST genes – how big is the eGST region itself and is it central to this? From Fig 6a it seems potentially so for the 1534C associated SNPs but much less so for the 1016G associated SNPs. What other genes are in this region? It might be helpful to report the SNPs in a supplementary table.

>>> We have rewritten this section to make clear that, while the 15 GSTs are overwhelmingly the most obviously resistance-associated genes in the sweep region, there is also an ankyrin-3 gene within this region that could have VSSC-associated functions. We had previously included the positions of the SNPs in Supp Table 7, but have reformatted this section to make this list more obvious to the reader.

“The region of SNPs on chromosome 2 strongly associated with VSSC genotype covered a 2,761,683 bp region. This region contained 32 SNPs strongly associated (adjusted P-values $< 1.956 \times 10^{-5}$) with the V1016G and F1534C mutations (Fig 6a; 13 and 19 SNPs respectively). Genes within this region included 15 glutathione S-transferase (GST) epsilon-class genes with obvious links to resistance 28–30, as well as an ankyrin-3 gene with possible VSSC-associated function 43,44. Positions of the 32 SNPs are listed in Table S7.”

Reviewer #2 (Remarks to the Author):

Dear Authors,

I recently had the opportunity to review your manuscript. I commend you on the comprehensive analysis and clear presentation of your research on insecticide resistance in *Aedes aegypti* and *Ae. albopictus*. The depth of your study is impressive, and it provides significant insights into the genetic mechanisms of resistance in these crucial vector species.

>>> We thank the Reviewer for their useful comments and their positive assessment of the MS.

I do, however, have a few queries and suggestions that I believe could further enhance the impact and utility of your work:

Imputation Methodology: Your decision to use Beagle for imputation piqued my interest. Could you elaborate on why you chose this method? Specifically, I'd like to know whether you utilized micro haplotypes for data input or relied solely on statistical imputation.

>>> We have rewritten this section in the Method to clarify the specific usage of Beagle v4.1:

“Missing data were imputed in windows of 6000 SNPs using Beagle v4.1, using 10 iterations, 500 SNP overlaps, and the full dataset as reference.”

We did not use microhaplotypes for imputation. We instead used sliding windows of SNPs across each contig to assist with imputation. We agree that a phased microhaplotype-based method could have advantages over the method we used, but overall this is unlikely to have any discernible effect on our results given that missing data was very low across both datasets (*Ae. aegypti*, $\bar{x} = 2.42\%$; *Ae. albopictus*, $\bar{x} = 4.88\%$). With such low levels of missing data, even no imputation (i.e. taking the mean genotype for missing data) gives similar results. For the sections of this study where the precise genotype is required, e.g. for identifying GST sweep haplotypes, or VSSC backgrounds, we did not use any imputation.

Linkage Network Analysis: Have you considered employing linkage network analysis, perhaps using LDna, to examine the genetic architecture in greater detail? Such an analysis could provide additional layers of understanding regarding the genetic relationships and dynamics within and between the studied populations.

>>> This is an excellent suggestion.

We conducted linkage network analysis individually on the six swept backgrounds in *Aedes aegypti* (F1534C Americas, F1534C Indo-Pacific, V1016G, GST Australia, GST Americas, GST Indo-Pacific). We did not use LDna, but instead ran a linkage network analysis for each of these six backgrounds via the following:

- 1) For each, group, we used VCFtools to calculate LD (r^2) between SNPs within 1mb of the relevant swept region (i.e. for GST sweeps it was within 1mb of the GST region) and SNPs more than 50mb away, including those on other chromosomes.
- 2) After trialling several cut-offs iteratively, we used $r^2 > 0.6$ as a cutoff for SNPs in high LD with the swept region.
- 3) We built histograms showing the density of these SNPs across the genome in 1 Mb bins.

The results from this are displayed in Fig 9 (previous Fig 9 is now Fig 10). For all of the sweeps apart from GST Australia, there were clusters of SNPs across the genome in high LD with the swept region that contained genes with known resistance function. Both F1534C sweeps were in strong linkage with the GST epsilon genes, but no GST epsilon sweeps were linked to the VSSC gene. However, the GST Americas and GST Indo-Pacific sweeps were both linked to a specific acetylcholinesterase (ace) gene on chromosome 3. Aside from these repeated observations, the general pattern observed was that each sweep was associated with a different set of resistance genes across the genome, including cytochrome P450s, GSTs, and esterase B1s.

These intriguing patterns suggest that, as a complement to the repeated evolution at the VSSC and GST epsilon gene regions, selection has been able to act on variation from a broad range of genes genome-wide, leading to a range of genetic architectures underpinning local resistance phenotypes. The observation that no resistance genes were associated with the GST Australia sweep adds further evidence to our hypothesis that this represents a second sweep back to a wild-type background.

We also ran the above analysis on *Aedes albopictus*, with results in Fig 10g. Here, we plotted only the most strongly associated contigs. We found similar patterns to those in *Aedes aegypti*, but the data were much 'noisier' overall, possibly due to low sample size ($n = 8$).

We have made relevant edits throughout the paper to incorporate this new analysis which adds considerably to the story.

Code Sharing and Reproducibility: I am thoroughly impressed with your analytical approach. To further benefit the scientific community, especially graduate students and researchers looking to replicate or build upon your work, share the detailed code of your analysis. Creating a step-by-step guide in a format like Quarto or Markdown, accompanied by all necessary files and raw data, would be immensely valuable.

Platforms like Zenodo or Dryad are excellent for depositing these materials. While I understand this is at your discretion, it would significantly contribute to the study's transparency and facilitate academic collaboration.

>>> We agree that sharing code is vital for transparency and understanding. Our code and requisite other files will be made available via Dryad repository, which will cover sequencing processing, analysis, and plotting.

In my review of your manuscript, I have included several suggestions directly in the Word document for your consideration. Please note that some of these suggestions reflect American English conventions, and you are welcome to disregard them if they do not align with your preferred style or dialect. I intend to enhance the clarity and readability of your manuscript while fully respecting your linguistic preferences and writing style.

>>> Thank you for this. We have incorporated most of these suggestions which have helped make the MS more concise and clearer.

Your work is undoubtedly a significant contribution to the field, and these suggestions aim to maximize its impact and accessibility. Thank you for the opportunity to review your manuscript. I look forward to its continued development and contribution to our understanding of mosquito-borne disease control.

>>> Thank you again for the positive and constructive review.

Reviewer #3 (Remarks to the Author):

Historically, GST is well known to be involved in DDT, organophosphate, and carbamate insecticide resistance. Some papers have reported the association between GST and pyrethroid resistance but to my best knowledge, no clear conclusion has been reached. Historically, both pyrethroids and organophosphates have been used to control *Aedes* mosquitoes and it is not surprising that genes associated with resistance to these two groups of insecticides are accumulating in mosquitoes. Most experts whose major is insecticide resistance will simply understand that two genes (regions) have been selected by use of organophosphates and pyrethroids. I think the authors should describe the importance of the paper more clearly.

>>> We thank the Reviewer for invaluable input into how we can make the importance of this study clearer to insecticide resistance researchers.

Unfortunately, in this paper, no evidence whether any of the 15 GSTs found in this study are involved in resistance to any insecticide are indicated. If the authors show the clear evidence that any of the GSTs confer pyrethroid resistance, I believe it is worth to be published in this journal. But without showing such evidential data, I believe this paper fails to present results that exceed the expectations of many insecticide resistance researchers.

>>> We thank the Reviewer for raising this point. We have rewritten parts of the MS to more clearly indicate its importance to the understanding of insecticide resistance.

This MS contains several important results, and the Reviewer is right to point out that a very important result concerns the GST epsilon genes. Here, our key finding is that there are three evolutionarily independent genetic backgrounds at this locus that have each spread via natural selection across multiple populations that are thousands to tens of thousands of kilometres apart. Here we provide further context to this finding.

In the current genome assembly for *Aedes aegypti*, there are 15 genomic regions that contain one or more GST genes, and 39 GST genes in total. The GST epsilon class is just one of these 15 regions, and contains 15 GST genes, though our copy number variation findings (Fig 8) suggest there may be more of these genes in North American populations and possibly fewer copies in the Middle East and South Asia. This is in stark contrast to the VSSC gene, of which only one is found in this species. What is thus immediately notable about our findings is that these three strongly-selected backgrounds have all evolved in the same set of GSTs, the epsilon class (point #1).

What is then also notable is that these backgrounds have undergone selective sweeps across populations that show similar patterns to sweeps at the VSSC gene (point #2). Point #2 provides critical insight into the role of these specific genes in producing insecticide resistance. This is because GSTs are believed to confer metabolic resistance, and metabolic resistance is operationally treated by the WHO as having polygenic genetic architectures (WHO 2012, cited in main text). Under this view, the 39 GSTs are each able to contribute as small-effect loci to resistance phenotypes along with the 148 cytochrome P450 genes and various other genes across the genome. However, our results strongly suggest that the three backgrounds that have evolved and spread at the GST epsilon locus function more like those at the VSSC locus, that is, as large effect loci that each have a major influence over a population's level of resistance.

The phenotypic effects of these GSTs do not appear to be due to copy number variation (point #3). While we found strong evidence of copy number variation in some regions, particularly North America, there was no evidence that individuals with sweep backgrounds had higher read depth ratios than wild types from the same region (Fig 8b). This finding is important as it suggests that the large phenotypic effects of the three GST backgrounds are not simply due to gene duplications, which if true would still allow individual genes to have small effects, with the large effect due to local increases in copy number. Instead, it appears more likely that the GST backgrounds provide resistance via some other mutational mechanism such as point mutations. A single GST epsilon point mutation was found to correlate with resistance in *Anopheles* mosquitoes in Benin (Riveron et al. 2014 *Genome Biology*), so this mechanism is plausible.

Following the suggestion of Reviewer 2, we investigated LD networks, and these indicate another level to our understanding of genetic architecture (point #4). Point #1 demonstrates that the GST epsilon class has been a specific site for repeated evolution of resistance, but LD network analysis identified a range of other GSTs as well as P450s and other metabolic resistance-associated genes that were strongly linked to the six sweep backgrounds (3 VSSC, 3 GST). Different sets of sweep-linked genes were identified for each sweep background. The same analysis run on each region showed that linked genes were mostly region-specific, a pattern consistent with these genes evolving locally but not spreading with the sweeps.

Together these points show that the GST epsilon region specifically has undergone repeated evolution of resistance (point #1); that these GST resistance backgrounds confer sufficient phenotypic advantages to sweep across distant populations, with a similar genomic signature to sweeps at the VSSC gene, and in contrast to the usual expectations of metabolic resistance genes (point #2); that, while copy number varied significantly among some populations, this did not appear to be driving the phenotypic advantage, which will be due to some other mutational mechanism (point #3); and that the GST and VSSC resistance backgrounds are linked to other resistance genes across the genome, including other GSTs, P450s, and ace genes, which have mostly local effects (point #4).

We think these findings are a major contribution to our understanding of insecticide resistance in *Aedes* and in general. From a management perspective, our results provide several key new insights. The first is that metabolic resistance can spread rapidly via gene flow from populations with GST sweep backgrounds, in a similar way to target-site resistance at the VSSC gene has. The second is that, unlike VSSC backgrounds, GST sweep backgrounds may be able to increase their phenotypic effects via gene duplication as well as other forms of mutation. The third is that, by identifying 32-SNP haplotypes for each GST sweep, we provide a means of tracking the spread of these resistance backgrounds, allowing for the future development of assays to screen for them and for copy number increases. The fourth is that these sweeps likely confer fitness costs when insecticide use is low, and thus reducing or rotating insecticides may be beneficial in limiting GST epsilon spread; this insight is derived from the fact that all GST sweeps were partial (i.e. segregating in their populations), and from the specific results of the GST Australia sweep (detailed in the Discussion). Having identified these sweeps in this study, we agree with the Reviewer that future research should aim to link specific variation in these swept backgrounds with resistance to specific insecticides.

We have made major changes to the main text in the Discussion to make these points.

The knockdown resistance mutations found in the voltage sensitive sodium channel (Vssc), the target site of pyrethroid insecticide, is well studied in *Aedes aegypti*. It is well known that V1016G and F1534C are widely distributed in the world. These are the major mutations in *Ae. aegypti* but some other mutations, like V253F and V410L, have also proven to be involved in pyrethroid resistance. In *Ae. albopictus*, F1534C is the major mutation that confers pyrethroid resistance but recently, V1016G has also been detected in several countries including Italy, Vietnam, China, Bulgaria, France, Georgia,

Malta, Romania, Spain, Switzerland, and Turkey. So the authors should show the genetic data of mosquitoes having these other kdr mutations.

>>> We agree that the V1016G mutation in *Aedes albopictus* is important to study and further genomic research should be conducted into its evolution. In this study, we screened all 490 *Aedes albopictus* for this mutation and found no evidence of it. However, with the exception of Vietnam, we did not have access to genomic data or raw DNA from the populations listed by the Reviewer. There are published ddRAD sequence data from these populations, but unfortunately the authors of those studies used a different set of restriction enzymes for digesting the DNA, and thus when we previously tried to incorporate these data in this study, there were only ~100 SNPs common to both datasets. Similarly, as we do not have raw DNA from these populations, we are unable to genotype them at specific VSSC mutations.

We agree that these populations are very important to understanding resistance in *Aedes albopictus*, however, our study contains a wide range of populations that are also very important and have yet to be investigated genomically. Our research groups are based in the Indo-Pacific region, where 70% of global dengue is transmitted (Bhatt et al. 2013 *Nature*), and our sampling reflects this geographical focus. In particular, this study includes 204 *Aedes albopictus* and 69 *Aedes aegypti* from Papua New Guinea, an area of high dengue incidence but with little information on mosquito genomics.

Line 356: This is wrong. There has been no haplotype having both L982W and V1016G.

>>> This has been changed. We thank the Reviewer for detecting this error.

Generally, purpose of the study is not clear. The last three lines of the summary should be more specific as to what is written, i.e., how clarification of the resistance background is useful for vector control.

>>> We have rewritten this section as well as other rewrites throughout in line with this comment and the comments above.

I think that the entire paper needs to be written in a more reader-friendly manner. For instance, the word “sweep” or “swept” is difficult to understand for general readers of the journal.

>>> We have clarified and simplified language around selective sweeps where possible. In particular, we now introduce terms more clearly in the Introduction.

Reviewers' Comments:

Reviewer #1:

Remarks to the Author:

The revision has satisfactorily addressed my queries and comments

Reviewer #2:

None

Reviewer #3:

Remarks to the Author:

The authors did not answer my first question: mutations in Vssc are involved in pyrethroid insecticide resistance, while GSTs are generally involved in organophosphate and carbamate resistance. And there are numerous isozymes of GSTs, not all of which detoxify insecticides. I understand that this paper shows that gene duplication of GST and kdr mutations have been occurring globally, but it does not show at all what insecticide resistance the GST gene duplication is involved in. It is my opinion that this paper should not be published without evidence to show the function of duplicated GSTs.

REVIEWERS' COMMENTS

Reviewer #3 (Remarks to the Author):

The authors did not answer my first question: mutations in Vssc are involved in pyrethroid insecticide resistance, while GSTs are generally involved in organophosphate and carbamate resistance. And there are numerous isozymes of GSTs, not all of which detoxify insecticides. I understand that this paper shows that gene duplication of GST and kdr mutations have been occurring globally, but it does not show at all what insecticide resistance the GST gene duplication is involved in. It is my opinion that this paper should not be published without evidence to show the function of duplicated GSTs.

>>> We thank the reviewer for their feedback on the manuscript. We have made edits throughout the Introduction and Discussion to clarify several points and highlight the main findings of the paper. We have clarified that it is not copy number variation that is driving the sweeps at the GST genes (Fig 8b), but some other mechanism such as point mutations, as seen in the VSSC gene and in GST epsilon genes in *Anopheles funestus* in Benin (Riveron et al., 2014 *Genome Biol*). There is also the possibility that the three GST sweep backgrounds confer resistance to different groups of insecticides, as observed in the two VSSC mutations F1534C and V1016G.

We strongly agree with the reviewer that it is important to determine which insecticide groups are resisted by the GST sweep backgrounds identified here. However, we think this is beyond the scope of this paper which already has a large scope and where the insights are based on samples from across many populations that have been sampled and sequenced incrementally over more than a decade. Future work should aim to sample specific populations identified through this study, isolate particular sweep backgrounds in the lab, and assess resistance through bioassays.